# Differential modification of the C-terminal tails of different α-tubulins and their importance for microtubule function in vivo

**Mengjing Bao, Ruth E Dörig, Paula Maria Vazquez-Pianzola, Dirk Beuchle, Beat Suter\***

Institute of Cell Biology, University of Bern, Bern, Switzerland

**Abstract** Microtubules (MTs) are built from α-/β-tubulin dimers and used as tracks by kinesin and dynein motors to transport a variety of cargos, such as mRNAs, proteins, and organelles, within the cell. Tubulins are subjected to several post-translational modifications (PTMs). Glutamylation is one of them, and it is responsible for adding one or more glutamic acid residues as branched peptide chains to the C-terminal tails of both α- and β-tubulin. However, very little is known about the specific modifications found on the different tubulin isotypes in vivo and the role of these PTMs in MT transport and other cellular processes in vivo. In this study, we found that in *Drosophila* ovaries, glutamylation of α-tubulin isotypes occurred clearly on the C-terminal ends of αTub84B and αTub84D (αTub84B/D). In contrast, the ovarian α-tubulin, αTub67C, is not glutamylated. The C-terminal ends of αTub84B/D are glutamylated at several glutamyl sidechains in various combinations. Drosophila *TTLL5* is required for the mono- and poly-glutamylation of ovarian αTub84B/D and with this for the proper localization of glutamylated microtubules. Similarly, the normal distribution of kinesin-1 in the germline relies on *TTLL5*. Next, two kinesin-1-dependent processes, the precise localization of Staufen and the fast, bidirectional ooplasmic streaming, depend on *TTLL5*, too, suggesting a causative pathway. In the nervous system, a mutation of *TTLL5* that inactivates its enzymatic activity decreases the pausing of anterograde axonal transport of mitochondria. Our results demonstrate in vivo roles of TTLL5 in differential glutamylation of α-tubulins and point to the in vivo importance of α-tubulin glutamylation for cellular functions involving microtubule transport.

**\*For correspondence:**
Beat.Suter@izb.unibe.ch

**Competing interest:** The authors declare that no competing interests exist.

## Editor's evaluation

Chemical modifications of the microtubule cytoskeleton regulate microtubule function in cells, but many open questions remain, such as the physiological relevance of these modifications and their impact on proteins that bind microtubules, such as molecular motors. In their useful study, the authors combine biochemistry, gene editing, and imaging to analyze one type of tubulin modification, tubulin glutamylation, which is enriched in developing ovaries and neurons. Their solid work implicates this modification in cytoplasmic streaming in oocytes and mitochondrial movement in neurons, which suggests that this modification may affect the activity of the molecular motor, kinesin-1. This work would be of interest to cellular and developmental biologists.

## Introduction

Microtubules (MTs) are fundamental cytoskeletal filaments comprising heterodimers of α- and β-tubulins. Despite their high level of conservation in eukaryotes, MTs are still quite diverse because cells

**eLife digest** Cells are brimming with many different proteins, compartments, and other cell components that all play specific roles, often at very precise locations in a cell at particular moments in time. Human cells, like those of other animals and plants, contain long tracks called microtubules that are able to transport such components to wherever they are needed.

Microtubules consist of chains of proteins known as tubulins that the cell can modify with small molecule tags at specific locations. For example, an enzyme called TTLL5 attaches molecules of glutamic acid to multiple positions on one of the tubulin proteins (known as α-tubulin). However, it remains unclear what role such modifications have on the ability of microtubules to move components around the cell.

Fruit flies are often used as models of animal biology in research studies. Three different versions of α-tubulin are found within the ovaries of fruit flies. Two of these are 'general' α-tubulins that are expressed in almost all tissues around the body, but the third is exclusively made in the ovaries.

Bao et al. studied the effect of TTLL5 activity on microtubules in fruit flies. The experiments revealed that TTLL5 played a crucial role in adding glutamic acid marks to the two general α-tubulin proteins. These modifications were needed for microtubules to successfully distribute a transporting motor protein named kinesin-1 to where it was needed for cargo transport within the egg cells. On the other hand, glutamic acid tags were not added to the oocyte α-tubulin protein.

Further experiments studied nerve cells, called neurons, in the wings of the flies. In mutant fruit flies with inactive TTLL5 enzymes, cell compartments known as mitochondria moved along microtubules to one end of the neurons with fewer pauses than those in normal cells.

This work shows that glutamic acid tags play important roles in regulating the transport of cell components along microtubules in fruit flies. In the future, these findings may support efforts to develop new treatments for human neurodegenerative diseases that are linked to defects in microtubules.

---

express different tubulin genes and proteins, which can undergo several different post-translational modifications (PTMs). Based mainly on in vitro assays, these variants are thought to optimize the MT interaction with different microtubule-associated proteins (MAPs) and motors.

*Drosophila melanogaster* possesses four α-tubulin genes. Three of them are expressed during oogenesis and early embryogenesis, suggesting that they contribute to the formation of microtubules during these stages. The three genes are *αTub84B*, *αTub84D*, and *αTub67C* (*Kalfayan and Wensink, 1982*). αTub84B and αTub84D differ in only two amino acids. Their primary sequence in the C-terminal domain is identical, and both are expressed in most tissues and throughout development. In contrast, αTub67C is expressed exclusively during oogenesis, where it is maternally loaded into the egg and embryo. The primary structure of αTub67C is distinctively different from αTub84B/D (*Theurkauf et al., 1986*). These differences are also apparent in the C-terminal domain and include the last residue phenylalanine (Phe, F) in αTub67C and tyrosine (Tyr, Y) in αTub84B/D. *αTub85E* is the fourth α-tubulin. It is mainly expressed in the testes but not in the ovaries (*Kalfayan and Wensink, 1982*).

MTs are subjected to several PTMs, including C-terminal tail glutamylation, glycylation, and detyrosination/tyrosination. Furthermore, acetylation and phosphorylation in the more central parts of the tubulins have also been identified (*Arce et al., 1975*; *Eddé et al., 1990*; *Hallak et al., 1977*; *L'Hernault and Rosenbaum, 1985*; *Magiera and Janke, 2014*; *Redeker et al., 1994*). Glutamylation is a PTM that adds one or more glutamic acids (Glu, E) to the sidechains of glutamic acid residues in the C-terminal tails of both α- and β- tubulin, causing the formation of a branched peptide chain (*Eddé et al., 1990*; *Redeker et al., 1992*). Glycylation is another PTM that adds glycines to the γ-carboxyl group of glutamic acid residues in the C-terminal region (*Redeker et al., 1994*). At least in vertebrates, tubulin is also subjected to a particular cycle of de-tyrosination-tyrosination (*Preston et al., 1979*). In this case, the C-terminal Tyr or Phe of α-tubulin is cyclically removed, resulting in a C-terminal Glu residue, and a new Tyr can be readded to the new C-term. The enzymes that catalyze glutamylation, glycylation, and tyrosination belong to the Tubulin-tyrosine-ligase-like (TTLL) family (*Janke et al., 2005*). In *D. melanogaster*, this family encompasses 11 genes (flybase. org).

**Table 1.** Conservation between mammalian TTLL5 and *Drosophila* TTLL5.

Residues labeled in red are critical residues for α-tubulin glutamylation by murine TTLL5 and are conserved in human and *Drosophila* TTLL5 (*van Dijk et al., 2007*; *Natarajan et al., 2017*).

| | K131 | β6-7 loop (R188) | R225 | E366 N368 |
|---|---|---|---|---|
| *Mm* TTLL5 | TRKDR--KP | VASSRGRG | VY---VRLYVL---LLEVNLSP | |
| *Hs* TTLL5 | TRKDR--KP | VASSRGRG | VY---VRLYVL---LLEVNLSP | |
| | K282 | β6-7 loop (R339) | R376 | E517 N519 |
| *Dm* TTLL5 | TRKDR--KP | AASSRGRG | IF---LRVYVL---LLEINLSP | |

The critical domain of the β6-7 loop is underlined.

MTs have an intrinsic polarity with a plus and a minus end. On these tracks, kinesin motors transport cargo to the plus ends, and dynein motors move cargo toward the minus ends. Evidence for the importance of PTMs of tubulin subunits has already been reported in vitro. Artificial tethering of 10E peptides to the sidechains at position E445 of α-tubulin and E435 of β-tubulin by maleimide chemistry increased the processivity (the run length) of kinesin motors on tubulin. For kinesin-1, this was a 1.5-fold increase (*Sirajuddin et al., 2014*). In contrast to kinesin, Dynein/dynactin motors preferred a Tyr at the C-term of α-tubulin to initiate processive transport in vitro because the motor favored this isotype as an attachment point to dock onto the MTs (*McKenney et al., 2016*). Recent work now also demonstrated a role for the mouse tubulin glycylation enzymes. In vitro fertility assays showed that the lack of both *TTLL3* and *TTLL8* perturbed sperm motility, causing male subfertility in mice. Structural analyses further suggested that loss of glycylation perturbed the coordination of axonemal dynein, thus affecting the flagellar beat (*Gadadhar et al., 2021*).

*CG31108* is considered to be the homolog of the mammalian *TTLL5* gene and was therefore named *DmTTLL5 or TTLL5* (*Devambez et al., 2017*). It is required for α-tubulin glutamylation in the *Drosophila* nervous system (*Devambez et al., 2017*). The murine TTLL5 is composed of an N-terminal core TTLL domain, a cofactor interaction domain (CID), and a C-terminal receptor interaction domain (RID) (*Lee et al., 2013*). The N-terminal core domain was shown to provide the catalytic activity of TTLL5 and is highly conserved in *Drosophila* TTLL5 (*van Dijk et al., 2007*, *Natarajan et al., 2017*; *Table 1*). According to FlyBase data, *Drosophila TTLL5* mRNA is expressed at high levels in ovaries (*Supplementary file 1*), even higher than in the nervous system where initial studies were performed (*Devambez et al., 2017*). We thus focused on *TTLL5*'s possible functions during *Drosophila* oogenesis to shed light on possible in vivo roles of glutamylated α-tubulins.

An in vivo mouse study showed that the absence of TTLL1-mediated polyglutamylation of α-tubulin increased the overall (both anterograde and retrograde) motility of mitochondrial transport in axons (*Bodakuntla et al., 2021*). However, another study in *Drosophila* showed that the vesicular axonal transport was not affected by the absence of TTLL5-mediated α-tubulin glutamylation in the segmental nerves of larvae (*Devambez et al., 2017*). Inspired by these contradictory results, we also studied the function of *TTLL5* in a second tissue, the adult *Drosophila* wing nerves.

Our results revealed that *TTLL5* is essential for the normal glutamylation of the ovarian αTub84B/D. Surprisingly, even in wild-type ovaries, the oogenesis-specific aTub67C was not glutamylated, suggesting that this gene, *αTub67C*, may have evolved to produce MTs that contain regions that are less glutamylated. We found that *TTLL5* is required for the proper distribution of Kinesin heavy chain (Khc), for proper cytoplasmic streaming during late oogenesis, as well as for fine-tuning the localization of the Staufen protein. Furthermore, *TTLL5* promotes the pausing of mitochondria during anterograde axonal transport in wing nerves. Our work, therefore, reveals the in vivo importance of glutamylation of specific α-tubulin isotypes.

## Results

### Specific glutamylation on multiple Glu residues of αTub84B/D, but not αTub67C in ovaries

Mass spectrometry was performed to analyze the glutamylation of the C-terminal tails of α-tubulins in vivo. Proteins extracted from ovaries were purified through SDS-PAGE, and α-tubulin bands were cut out and digested with the protease trypsin-N. C-terminal peptides of the different α-tubulin

**Table 2.** C-terminal peptides of the *Drosophila* α-tubulins analyzed by mass spectrometry (MS).

**(A) Number of C-terminal peptides of αTub84B, αTub84D, and αTub67C**

| Isotype | w | $TTLL5^{pBac/-}$ (PSM value*) | $TTLL5^{MiEx/-}$ |
|---|---|---|---|
| αTub84B/D | 58 | 24 | 6 |
| αTub67C | 12 | 12 | 4 |

**(B) Role of *TTLL5* for the glutamylation of the C-terminal E443, 445, 448, and 449 of αTub84B/D. Frequency of glutamylated C-terminal peptides (containing the entire primary sequence) in w controls, $TTLL5^{pBac/-}$, and $TTLL5^{MiEx/-}$ mutants containing modifications with the Glu sidechain length of 1E, 2E, or 3Es.**

| Sidechain length | w | $TTLL5^{pBac/-}$ (%E mod†) | $TTLL5^{MiEx/-}$ |
|---|---|---|---|
| +1 | 50 | 4 | 0 |
| +2 | 14 | 0 | 0 |
| +3 | 7 | 0 | 0 |
| Total Glu modifications‡ | 71 | 0 | 0 |

*The total C-terminal peptide numbers were revealed by the values of total peptide spectrum match (PSM) of peptides containing the unmodified full primary C-terminal sequence (**Supplementary file 1**) and the C-terminal sequence modified with the Glu sidechain (**Supplementary file 1**).

†The frequency was quantified based on the glutamylated C-terminal αTub84B/D PSM values (quantified based on **Supplementary file 1**) divided by total PSM for the C-terminal peptide. The total PSM for w, $TTLL5^{pBac/-}$, and $TTLL5^{MiEx/-}$ were 58, 24, and 6, respectively.

‡Total Glu modifications show the sum of the three frequencies.

isotypes were then analyzed for post-translational modifications by liquid chromatography-mass spectrometry (LC-MS). Because the resulting C-terminal peptides are identical between the αTub84B and αTub84D, we refer to them as αTub84B/D. According to FlyBase, *αTub84B+D* mRNAs are 4.7× as highly expressed as *αTub67C*. We identified a similar ratio (4.6×) of total C-terminal peptides by the MS analysis (*Table 2A*, *Supplementary file 1*). Analyses of wild-type ovarian extracts revealed that the general *Drosophila* α-tubulins αTub84B/D, but not the ovarian-specific αTub67C, showed glutamylation on their C-terminal tails (*Supplementary file 1*). For αTub84B/D, sidechain glutamylation was identified on all four glutamyl residues in the C-terminal tail region, Glu443, Glu445, Glu448, and Glu449 (*Figure 1*, *Supplementary file 1*). The most extended sidechains identified were 3Es in wild-type ovaries (*Figure 1B*, *Supplementary file 1*). Due to the lower abundance of Tub67C C-terminal peptides, we cannot exclude the possibility of rare glutamylation on the maternal αTub67C. Altogether, we conclude that the majority of C-terminal peptides of αTub84B/D are glutamylated in contrast to the C-terminal tail of αTub67C.

## *Drosophila TTLL5* is required for MT mono- and polyglutamylation of ovarian αTub84B/D

To study the function of TTLL5, distinct *TTLL5* loss-of-function mutations were created and characterized (*Figure 2A*). The $TTLL5^{pBac}$ and $TTLL5^{Mi}$ mutant strains carry a piggyBac and a Minos transposon, respectively, within the open-reading frame of *TTLL5*. The $TTLL5^{MiEx}$ mutant was generated by imprecise excision of the Minos element, which caused the formation of a premature stop codon at gene position 8132 in the open-reading frame (codon position 392). All *TTLL5* alleles were hemizygously crossed over *Df(3R)BSC679*, which removes the region of the *TTLL5* gene. An anti-TTLL5 antibody that cross-reacts with additional proteins in ovarian extracts also recognized a band corresponding to the calculated mass of TTLL5. This band was absent in ovarian extracts prepared from the three *TTLL5* mutants, suggesting that they are protein nulls for *TTLL5* or might make only a truncated protein (*Figure 2B*). A transgene containing the wild-type *Drosophila TTLL5* sequence fused to the Venus fluorescent protein sequence under UAS control (UASP-*Venus::TTLL5*) was also produced as a tool to rescue the phenotypes produced by the *TTLL5* mutations and for overexpression studies (*Figure 2A*).

To quantify the effect of *TTLL5* on the glutamylation of the C-terminal tails of α-tubulins, we determined the frequency of C-terminal peptides containing the entire primary sequence and 0, 1, 2, or 3

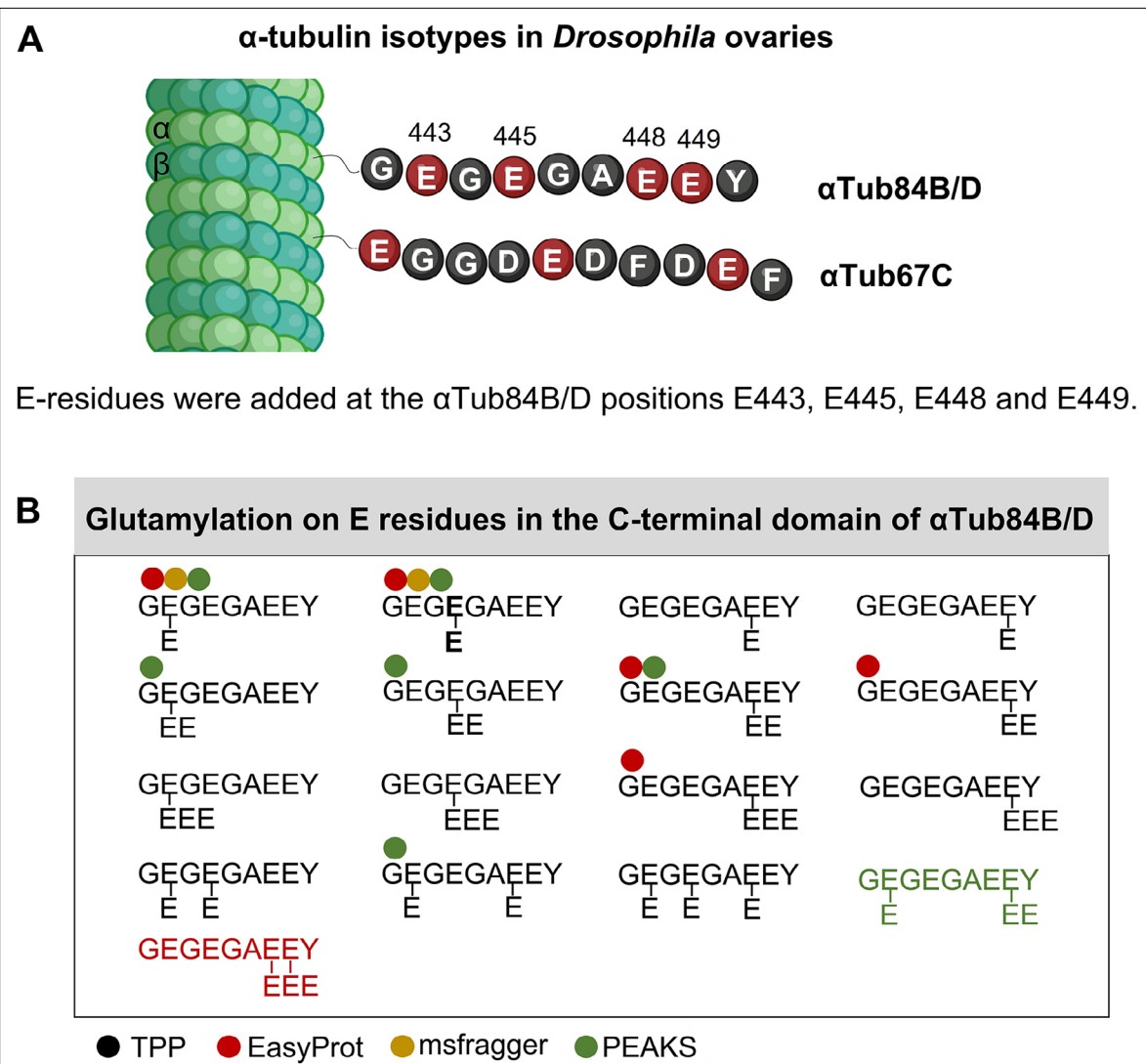

**Figure 1.** Mass spectrometry analysis of glutamylation of the C-terminal domains of *Drosophila* α-tubulins. (**A**) Schematic representation showing how the C-terminal regions of αTub84B/D can be modified by glutamic acid residues added to the sidechains of E443, E445, E448, and E449 in *Drosophila* ovaries. (**B**) The C-terminal peptides containing the complete primary sequence were analyzed. The structure of the different C-terminal tail peptides of αTub84B/D modified by glutamylation is shown starting with G442. Different combinations of Glu (E) sidechain modifications added at positions E443, E445, E448, and E449 of αTub84B/D were found. The glutamylation patterns were determined by the Trans-Proteomics Pipeline (TPP; *Supplementary file 1*), and the positions with the highest probabilities for E sidechain modification according to TPP are shown with black typeface. The 'E' in bold typeface indicates the most frequent position for mono-Glu modification identified in this study. Peptides identified by EasyProt, MSfragger, and PEAKS, respectively, are marked with a red, yellow, and green dot, respectively. These peptide identifications are shown in *Supplementary file 1*. Modifications only identified by EasyProt or PEAKS are displayed in the respective font color. Scheme was drawn by biorender.com.

Glu modifications. This analysis was performed with extracts from wild-type ovaries and the different *TTLL5* mutant ovaries. Wild-type ovaries revealed that 71% of the peptides were modified, 50% with 1E, 14% with 2Es, and 7% with 3Es (*Table 2B*). On the other hand, 4% (1 out of 24) mono-glutamylation was observed in *TTLL5^{pBac/-}* and no glutamylation in *TTLL5^{MiEx/-}* mutants (*Table 2B* and *Supplementary file 1*). This clearly shows that *Drosophila TTLL5* is needed for the sidechain glutamylation of the C-terminal Glu443, Glu445, Glu448, and Glu449 of αTub84B/D, and it appears to affect already the addition of the first Glu. The single monoglutamylated peptide found in the *TTLL5^{pBac/-}* mutant might point to another enzyme that has this activity or it might indicate that the *TTLL5^{pBac}* allele is not a true null allele. Additionally, this single peptide might also be a minor contamination from the isolation of the material or the MS analysis.

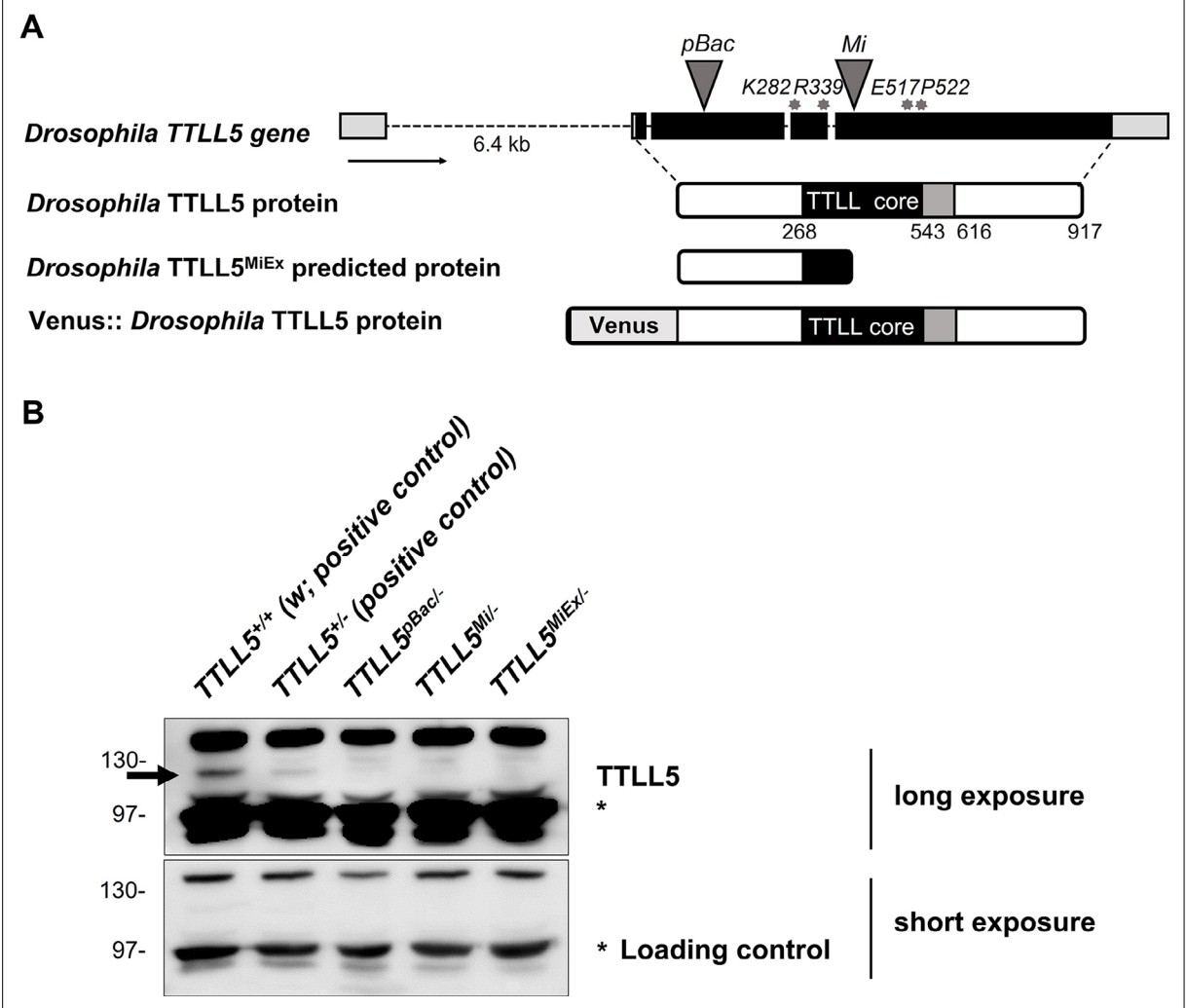

**Figure 2.** Gene structure and protein expression of the *Drosophila TTLL5* alleles. (**A**) The *TTLL5* gene structure is based on the FlyBase data for the CG31108-RA transcript. The positions of the loss-of-function mutations are marked on the *TTLL5* gene. Gray triangles: transposon insertions. Gray asterisks: codons targeted for generating InDels and point mutations. All *TTLL5* alleles were analyzed as hemizygous animals over *Df(3R)BSC679*, which removes the region of the *TTLL5* gene. (**B**) Western blot from the soluble fraction of ovarian extracts. w (*TTLL5⁺ᐟ⁺*) contains two wild-type copies of *TTLL5* and expressed the highest levels of TTLL5. The TTLL5 signal is strongly reduced or abolished in the null mutants *TTLL5ᵖᴮᵃᶜᐟ⁻*, *TTLL5ᴹⁱᐟ⁻* and TTLL5ᴹⁱᴱˣᐟ⁻. The hemizygous TTLL5⁺ ovaries (*TTLL5⁺ᐟ⁻*) expressed less TTLL5 than w. The signal of an unspecific band (labeled with *), produced after a short exposure, was used as the loading control. In the following, we will refer to the genotypes as w for the control (*TTLL5⁺ᐟ⁺*) and TTLL5ᵃˡˡᵉˡᵉ for the hemizygous mutants (*TTLL5ᵃˡˡᵉˡᵉᐟ⁻*).

The online version of this article includes the following source data for figure 2:

**Source data 1.** Source data for *Figure 2*.

We measured the levels of monoglutamylation of α-tubulin by western blotting using the GT335 antibody. This antibody reacts with the first glutamate on the glutamate sidechain even if it is polyglutamylated (*Figure 3A*; *Wolff et al., 1992*). On western blots from wild-type extracts, the GT335 antibody produced several bands. Bands pointed out by the arrowheads are at the expected position. In the mutants, the signal in the corresponding region is strongly reduced or absent but reappears in the rescued ovary extracts (*Figure 3B*). This result, therefore, supports the finding that *TTLL5* is required for monoglutamylation of α-tubulin, and it shows that the antibody signal is specific and can be used to monitor *Drosophila TTLL5* activity. Note that the GT335 antibody preferentially recognizes the first Glu attached to a Glu residue in an acidic environment. In mice, the middle Glu in the sequence -G**E**E- of the C-terminal tail of α-tubulin is such an 'acidic' site and the branched Glu modified on this residue can be identified by GT335 (*Bodakuntla et al., 2021*). In contrast, in *Drosophila* α-tubulin, both E443

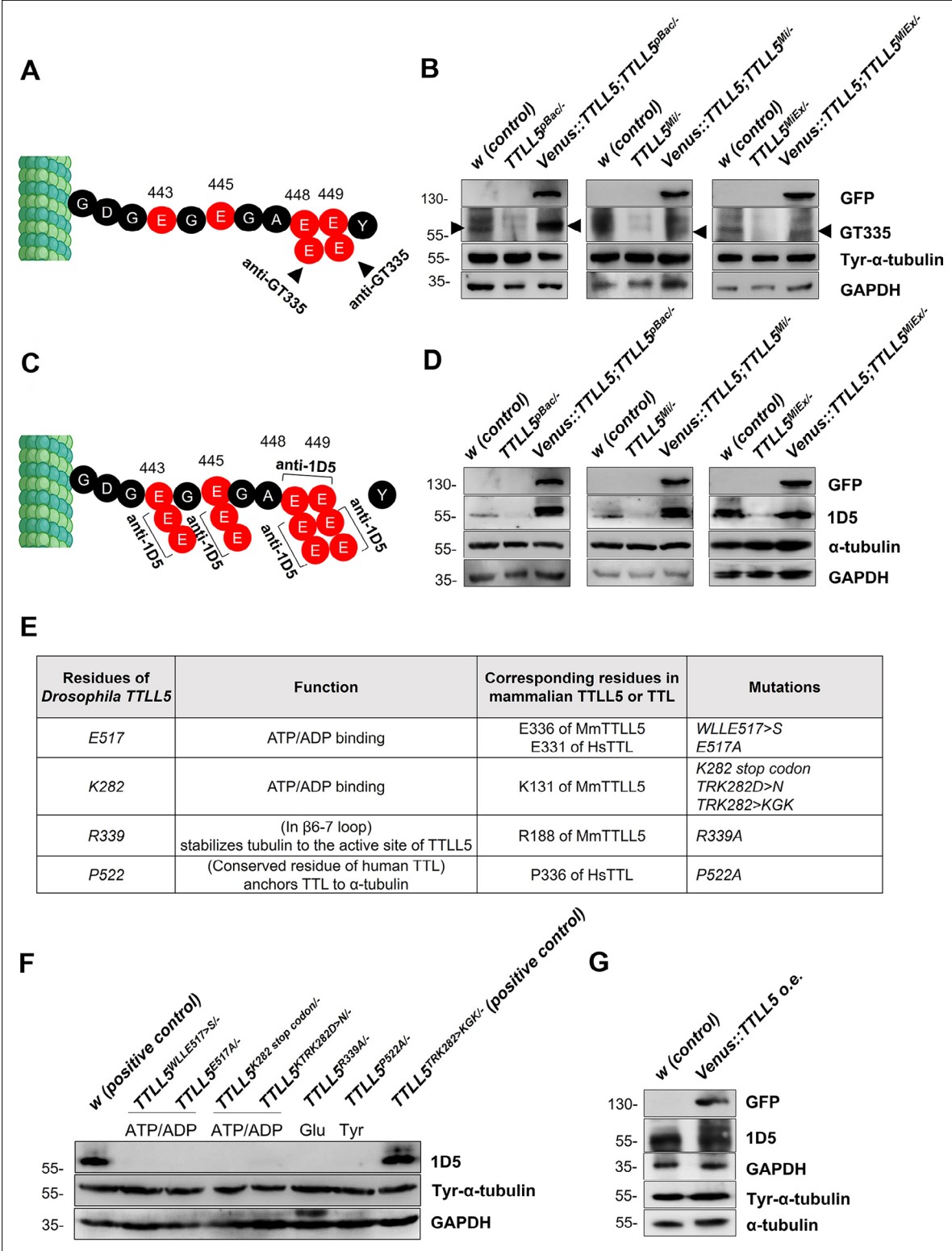

**Figure 3.** *TTLL5* is required for the glutamylation of α-tubulin in ovaries. (**A**, **C**) Sites on αTub84B/D expected to be recognized by the antibody against monoglutamylated α-Tubulin (GT335; **A**) and by the 1D5 antibody recognizing polyglutamylated αTub (**C**). (**B**, **D**) The glutamylation signal, produced by western blotting with the GT335 and 1D5 antibodies, respectively, is lower in TTLL5ᵖᴮᵃᶜ/⁻, TTLL5ᴹⁱ/⁻, and *TTLL5ᴹⁱᴱˣ/⁻* than in the *w* control but is restored and even elevated in mutant ovaries that express *Venus::TTLL5* under *MattubGal4* control. The upshifted bands recognized by 1D5 can be seen in the rescued TTLL5ᵖᴮᵃᶜ/⁻ and *TTLL5ᴹⁱ/⁻* mutants. The total tyrosinated α-tubulin levels seemed unaffected by the TTLL5 levels. α-tubulin and GAPDH served as loading controls. (**E**) The genotypes and rationale of the *TTLL5* alleles generated by CRISPR/Cas9 are listed. The selected residues are predicted to

*Figure 3 continued on next page*

*Figure 3 continued*

be either important for the glutamylation or tyrosination function based on the alignment shown in *Tables 1 and 3*. (**F**) Western blotting shows that the polyglutamylation signal is absent in all point mutants. *w* with two copies of *TTLL5⁺* is the wild-type control and the hemizygous TTLL5^TRK282>KGK282/-^ still contains the crucial K282 residue and behaves like the control. Relative to the loading controls, 1D5 signals decrease slightly in the hemizygous situation with only one *TTLL5⁺* copy. (**G**) Stronger upshifted bands were observed with the polyglutamylation antibody (1D5) upon *Venus::TTLL5* overexpression in wild-type ovaries. Total α-tubulin levels were similar in samples with excessive expression of *TTLL5*. GAPDH was a loading control. All mutant *TTLL5* alleles were analyzed as hemizygous animals over *Df(3)BSC679*. The genotypes of the rescued animals were *MattubGal4>UAS-Venus::TTLL5;TTLL5^alleles^/* Df(3R)BSC679. The genotype of the Venus::TTLL5 overexpressing animals was *MattubGal4>UAS-Venus::TTLL5*.

The online version of this article includes the following source data for figure 3:

**Source data 1.** Source data for *Figure 3*.

and E445 in the sequence -G**E**G**E**G- are flanked by Glycines on either side, which lacks the necessary acidic environment for efficient recognition by GT335. Consequently, the GT335 signal primarily arises from the glutamylation on the consecutive Glu residues E448 and E449 in the sequence -GA**EE**Y.

The 1D5 antibody recognizes α-tubulin with an -XEE sequence at the C-terminal α-carboxylate group (with X being a variable residue) (*Rüdiger et al., 1999*; *Figure 3C*). Therefore, in principle, the Glu sidechain on all residues E443, E445, E448, E449, and even the C-terminal residues E448/449 themself can be recognized if they are not followed by another amino acid (such as Tyr) at their α-carboxylate group. In our results, the control ovaries displayed high cross-reactivity of 1D5 with a polypeptide of the predicted size. This observation indicates the presence of glutamate sidechains containing at least two glutamates, as well as the possibility of C-terminally de-tyrosinated α-tubulin (*Gadadhar et al., 2017*; *Preston et al., 1979*; *Figure 3D*). Consistent with the MS analysis, the *TTLL5* mutants showed a strongly reduced signal with 1D5, indicating that loss of *TTLL5* function prevents normal polyglutamylation. However, the 1D5 signal was again restored in all *TTLL5* mutants upon expression of the rescue construct. Overexpression of TTLL5 in *TTLL5^pBac/-^* and *TTLL5^Mi/-^* even resulted in a stronger 1D5 signal and an upshifted band, suggesting the occurrence of hyperglutamylation on α-tubulin (*Figure 3D*). These findings not only demonstrate the important role of TTLL5 in facilitating the glutamylation of α-tubulin but also suggest that the endogenous levels of TTLL5 are typically limited in ovaries under normal conditions.

Most of the murine TTLL5 residues involved in α-tubulin glutamylation are conserved in *Drosophila* TTLL5 (*Table 1*; *Natarajan et al., 2017*). Two are known to be essential for the general enzymatic activity of TTLLs, and two for the interaction with the C-terminal domain of α-tubulin (*Figure 3E*). To determine whether these residues are indeed needed in vivo for the enzymatic activity of *Drosophila TTLL5* toward αTub84B/D, we used CRISPR/Cas9 to mutate the conserved codons in *Drosophila TTLL5*. *Drosophila* TTLL5 residues K282 and E517 correspond to mouse K131 and E336, respectively. They directly interact with ADP/ATP and thus affect general TTLL5 enzymatic activity (*Natarajan et al., 2017*). TTLL5^R339^ corresponds to mouse TTLL5^R188^, which is vital for TTLL5's glutamylation activity. MmTTLL5^R188^ resides in the β6–7 loop, which forms a salt-bridge interaction with the C-terminal tail of α-tubulin and orients it toward the ATP/ADP-binding site (*van Dijk et al., 2007*). Western blotting results showed that the glutamylation of α-tubulin was impaired in all these *TTLL5* mutants where the conserved residue was replaced (*Figure 3F*). Since these mutations did not abolish the stability of the corresponding mutant protein (*Appendix 1—figure 1*), our results revealed the importance of E517, K282, and R399 of *Drosophila* TTLL5 for glutamylation. Most active site residues of human TTL (*Prota*

**Table 3.** Conservation between TTL and TTLL5.
Many critical residues of human TTL (red) are conserved in *Drosophila*, mouse, and human TTLL5 (*Prota et al., 2013*).

| | R202 | R222 | D318 | E331 N333 P336 |
|---|---|---|---|---|
| *Hs TTL* | IRSW---VLRTA | | -----FDFM--- | LIEVNGAPA |
| | R376 | R398 | D504 | E517 N519 P522 |
| *Dm TTLL5* | LRVY---IVRLA | | -----FDIL--- | LLEINLSPS |
| | R225 | R247 | D353 | E366 N368 P371 |
| *Mm TTLL5* | VRLY---LARFA | | -----FDVL--- | LLEVNLSPS |
| | R225 | R247 | D353 | E366 N368 P371 |
| *Hs TTLL5* | VRLY---LARFA | | -----FDVL--- | LLEVNLSPS |

*et al., 2013*; *van Dijk et al., 2007*) are also conserved between mammalian and *Drosophila* TTLL5 (*Table 3*). We also mutated the *Drosophila* TTLL5$^{P522}$ codon, which corresponds to human *TTL*$^{P336}$ (and human *TTLL5*$^{P371}$/mouse *TTLL5*$^{P371}$). This residue is required for anchoring TTL to α-tubulin by forming hydrogen bonds with C-terminal tail residues of α-tubulin (*Prota et al., 2013*). Even though the importance of *TTLL5*$^{P371}$ for glutamylation has not been reported in mammalian systems, its replacement prevented glutamylation of α-tubulin in *Drosophila* ovaries. This identifies P522 as a novel critical residue not only for the tubulin tyrosination by TTL but also for the glutamylation function of TTLL5.

We also tested whether the overexpression of *TTLL5* in a wild-type background affected polyglutamylation similarly. Indeed, *UASP-Venus::TTLL5* overexpression in ovaries led to hyperglutamylation, as evident from the stronger signal and the additional upshifted band (*Figure 3G*). All these results point to *TTLL5's* essential role in the glutamylation of α-tubulin.

## Other PTMs of the C-terminal tail of ovarian α-tubulin

Western blotting revealed minimal or no alterations in the levels of tyrosinated α-tubulin in any of the null or CRISPR/Cas9 generated point mutants of *TTLL5*. Additionally, the overexpression of *TTLL5* did not affect these levels either (*Figure 3B, F and G*). Thus, the levels of TTLL5 do not appear to affect the levels of tyrosinated α-tubulin in ovaries.

The MS results from the ovarian samples contained low signals that could correspond to glycylated C-terminal α-tubulin peptides based on their molecular mass. Such signals were observed in αTub84B/D but not in αTub67C. However, the standard procedure for the MS analysis involves the alkylation of Cysteine residues in the sample and this procedure can also modify Glu residues, giving rise to the exact same mass change as glycylation (+57.021464 Da; *Kim et al., 2016*). To independently test for this PTM, we also performed western blotting experiments with ovarian extracts and probed them with the anti-mono Gly antibody TAP952 (*Bré et al., 1996*). However, even in the wild-type control, the anti-mono Gly antibody TAP952 did not reveal a clear signal at the predicted position (*Appendix 1—figure 2*), suggesting that α-tubulin is not glycylated in *Drosophila* ovaries. Furthermore, considering that the known *Drosophila* polyglycylases (TTLL3A/B) were shown to be the main glycylases in the whole fly and are not expressed in ovaries (*Rogowski et al., 2009*; *Supplementary file 1*), the current evidence does not justify further analysis of glycylation in *Drosophila* ovaries.

## Presence of polyglutamylated microtubules in ovaries

We then analyzed the distribution of polyglutamylated microtubules during oogenesis by immunofluorescence using the 1D5 antibody. Specific 1D5 immunofluorescence signals were observed in the follicle cells and oocytes throughout the previtellogenic, middle, and late oogenesis stages (*Figure 4*). In stage 10B (S10B) oocytes, the 1D5 signal showed a slightly biased localization in the cortical region of the oocyte. We observed a robust reduction of the 1D5 signal in *TTLL5* mutant ovaries, revealing that *TTLL5* is essential for the appearance of polyglutamylated microtubules in the ovaries.

## Effect of *TTLL5* on the refinement of posterior localization of Staufen

*osk* mRNA localization, combined with translation control, is essential for targeting Osk protein expression to its proper posterior compartment in the cell during mid and late oogenesis (*Weil, 2014*). Staufen is a protein that associates with *osk* mRNA into a ribonucleoprotein (RNP) complex during *osk* mRNA localization. It therefore also serves as a proxy for *osk* mRNA localization. Staufen/*osk* mRNA localization requires the dynein/dynactin-mediated minus-end transport machinery in the early stages of oogenesis. Subsequently, the Staufen/*osk* mRNA becomes localized to the posterior of the oocyte from the mid to late stages of oogenesis. This localization phase depends on kinesin-driven processes, which include active transport and ooplasmic streaming in S10B oocytes (*Kato and Nakamura, 2012*; *Lu et al., 2018*; *Brendza et al., 2000*; *Zimyanin et al., 2008*).

To determine whether *TTLL5* contributes to these processes, we studied the role of *TTLL5* in the localization of Staufen in ovaries lacking functional TTLL5. In the wild type, the localization of Staufen appeared very tight on the cortex with little lateral extension in S10B (*Figure 5A*). In ovaries lacking functional TTLL5, the Staufen localization appeared more diffuse in the S10B egg chambers of all *TTLL5* null mutants (*Figure 5B*). With variations between samples, this phenotype was also seen in Z-stacks. The broader expansion of the Staufen crescent was fully reversed in all UAS-*Venus::TTLL5* rescued strains (*Figure 5C*). These results show that *TTLL5* is required for tight cortical accumulation

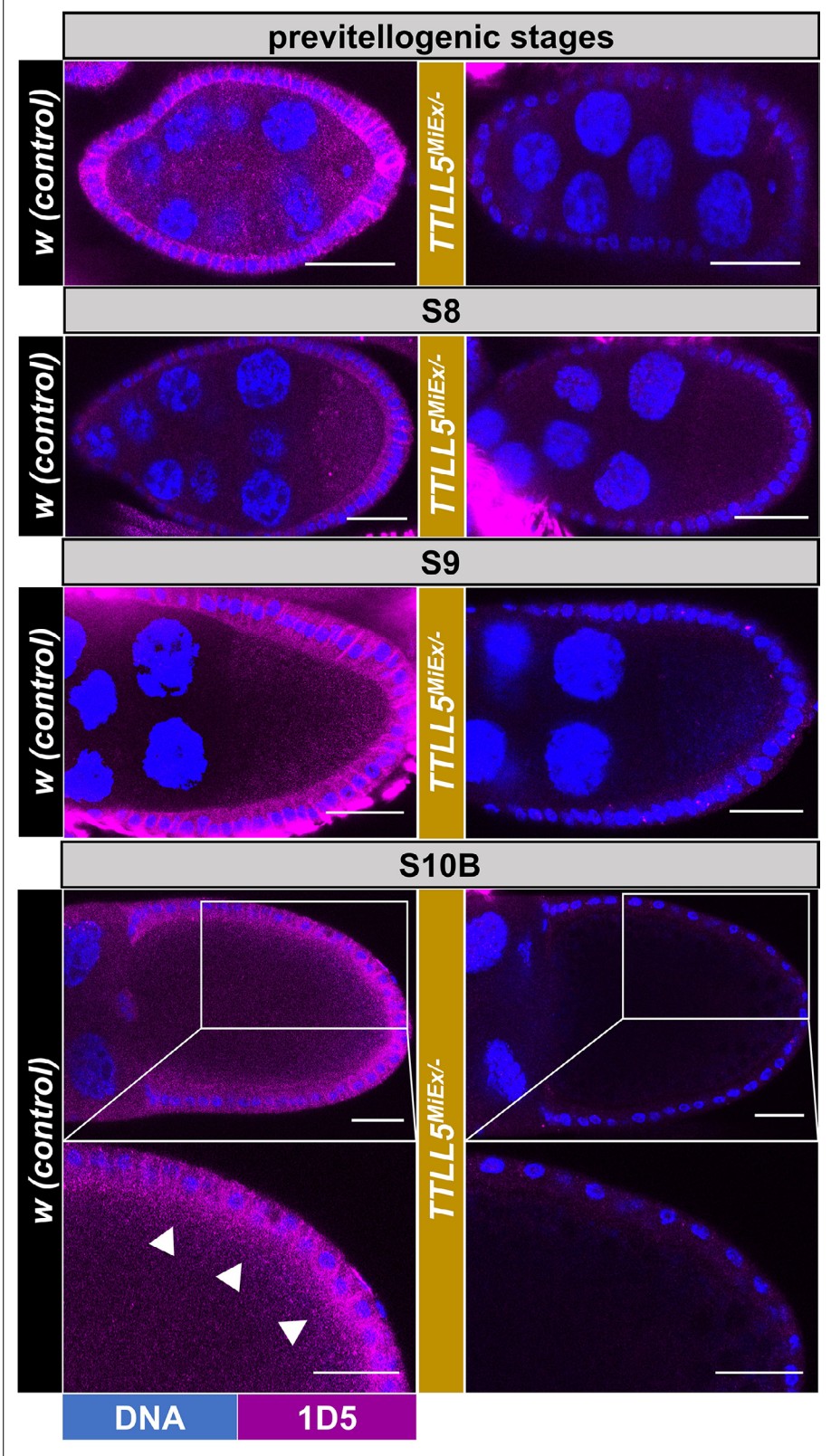

**Figure 4.** Spatial distribution of polyglutamylated microtubules in oocytes. Confocal micrographs showing representative oocytes and follicle cells from early to late stages of *w* controls and *TTLL5^MiEx/-^* mutants. Polyglutamylation (1D5) signals are shown in magenta and Hoechst (blue) stains the DNA. A 1D5 signal is seen in follicle cells and oocytes from previtellogenic to late-stage oocytes. For the S10B oocytes, a high-power

*Figure 4 continued on next page*

*Figure 4 continued*

magnification of the posterior half is shown, too. At this stage, a slightly biased 1D5 signal intensity was seen in the cortical region of the oocyte (marked by white arrowheads). The 1D5 signal is virtually absent in the *TTLL5* mutants. Imaging conditions and confocal microscope settings were identical for the two genotypes. The genotype of the control was *w*. Genotypes for *TTLL5^MiEx/-^* were *TTLL5^MiEx^/Df(3R)BSC679*.

of Staufen in S10 oocytes. Because the rescue construct was only expressed in the germline, they also strongly suggest that this requirement is a cell-autonomous one. We also analyzed the distribution of Venus::TTLL5 during oogenesis. Antibody staining suggested that Venus::TTLL5 strongly accumulated in the oocyte, especially along the cortex (*Figure 5C*). To validate these findings and rule out potential artifacts from antibody staining, we also attempted to monitor the self-fluorescent of the Venus tag. Although the Venus fluorescence was relatively weak, Venus::TTLL5 exhibited a cortical preference in S10B oocytes, similar to that found by antibody staining (*Appendix 1—figure 3*). This stronger cortical accumulation of Venus::TTLL5 coincides also with the region where polyglutamylated microtubules were enriched at the same stage (*Figure 4*).

In the *TTLL5* point mutants, where specifically the enzymatic activity of TTLL5 was blocked (*TTLL5^E517A/-^*, *TTLL5^P522A/-^*, *TTLL5^R339A/-^*) the localization pattern of Staufen was not refined and the Staufen signal remained less tightly focused (*Figure 5D*). An established method to quantify the localization of Staufen to the posterior cortex is to measure the length of the posterior Staufen crescent along the cortex (*Lu et al., 2016*). A shorter crescent is a sign of tighter localization. Indeed, we found that the length of the posterior Staufen crescent was significantly increased in *TTLL5* null and *TTLL5* point mutant ovaries compared to the control and UAS-*Venus::TTLL5* rescued ovaries (*Figure 5E*).

We did not observe clear differences in Staufen distribution between the wild type and *TTLL5* mutants in S9 and earlier, previtellogenic stages of oogenesis (*Figure 5F*). Altogether, these results suggest that the requirement for *TTLL5* for tight cortical accumulation of Staufen starts around S10 and the proper refinement of Staufen localization requires glutamylation of α-tubulin.

## The onset of fast ooplasmic streaming requires *TTLL5*

Ooplasmic streaming is a process that contributes to the refinement of the posterior localization of Staufen (*Lu et al., 2016*). It is a bulk movement of the oocyte cytoplasm that circulates and distributes mRNAs and proteins in the oocyte. In this way, Staufen/*osk* mRNA RNPs reach their intended position and become tightly anchored at the posterior end. Ooplasmic streaming can be observed in mid-stage to the late-stage oocytes. Slow and nondirectional flows initiate at stages 8–9 and are followed by rapid and circular streaming at S10B (*Lu et al., 2016*). The onset of the fast streaming phase has so far been reported to be attributed to (1) microtubule sliding between stably cortically anchored microtubules and free cytoplasmic microtubules close to the cortex (*Lu et al., 2016*) and (2) subcortical dynamic microtubule-mediated cargo transport (*Monteith et al., 2016*). Because the final posterior localization refinement of Staufen in S10B oocytes depends on the fast streaming (*Lu et al., 2016*) and also *TTLL5* (*Figure 5*), we wanted to determine whether *TTLL5* contributes to fast ooplasmic streaming.

We followed the streaming flow in real time by measuring vesicle movement in DIC time-lapse movies (*Videos 1–5*). We then tracked the movement of the vesicles along the posterior cortical region where microtubule sliding and transport occur (*Figure 6A*). The kymographs of the tracked vesicles, their analysis, quantification, and interpretation, are shown in *Figure 6B–D*. All control ovaries had a typical streaming pattern, and the vesicles were moving unidirectionally parallel to the cortex with a streaming center in the central part of the oocyte. However, 76% *TTLL5^MiEx/-^* and all *TTLL5^E517A/-^* mutant ovaries displayed abnormal streaming patterns. These abnormal streaming patterns included a partial disruption of streaming especially in the posterior part of the oocyte and a disordered streaming flow in the entire oocyte. The typical streaming pattern was restored in 71% *TTLL5^MiEx/-^* ovaries expressing a *Venus::TTLL5* rescue construct. We also evaluated streaming flows in oocytes overexpressing *Venus::TTLL5*. Elevated TTLL5 levels seemed to affect the streaming, too. 29% of the oocytes displayed an abnormal streaming movement or a weak streaming flow like the one observed in *TTLL5* mutants. Moreover, even in oocytes with a circular streaming pattern, the streaming center frequently appeared at a more anterior position in the oocyte, while the cytoplasmic flow in the posterior region was less clearly directed (see *Figure 6B*, #2, and *Video 5*, Sample 2). We

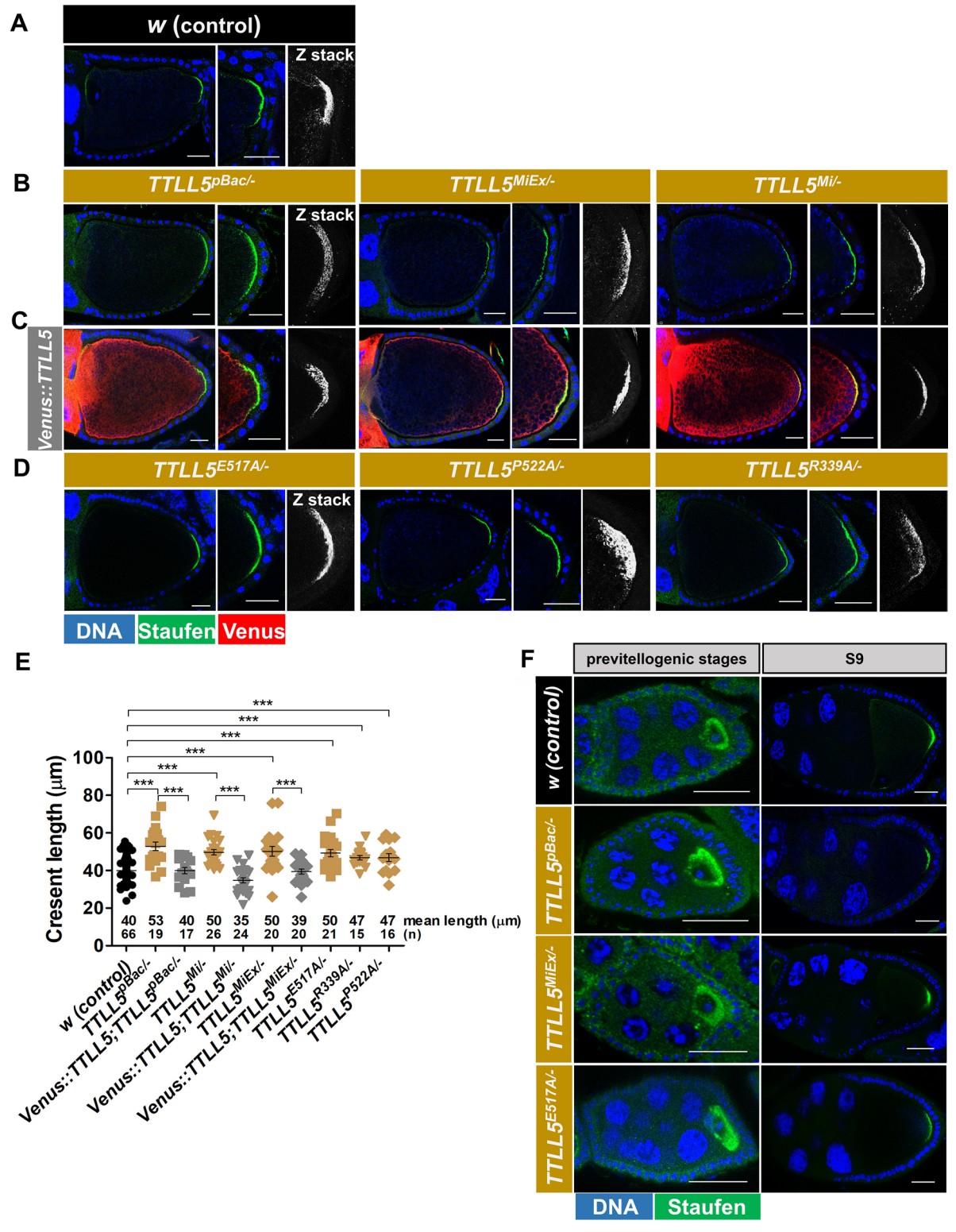

**Figure 5.** Effect of *TTLL5* on Staufen localization refinement in oocytes. (**A–D**) Confocal micrographs showing S10B oocytes of *w* controls (**A**), *TTLL5^pBac/-^*, *TTLL5^Mi/-^*, and *TTLL5^MiEx/-^* mutants (**B**), *TTLL5^pBac/-^*, *TTLL5^Mi/-^*, and *TTLL5^MiEx/-^* mutants rescued by *Mattub4>UASP-Venus::TTLL5* (**C**), *TTLL5^E517A/-^*, *TTLL5^P522A/-^*, and *TTLL5^R339A/-^* mutants (**D**). Anti-Staufen is shown in green, anti-GFP in red, and Hoechst (DNA) in blue. (**E**) Quantification of the Staufen crescent length along the posterior cortex based on (**A–D**) ***p<0.005. z-stack: 12 μm. (**F**) Confocal micrographs showing previtellogenic and S9 egg chambers of *w* controls, *TTLL5^pBac/-^*, *TTLL5^MiEx/-^*, and *TTLL5^E517A/-^* mutants. Staufen signal is shown in green, and Hoechst in blue. Scale Bar: 25 μm. *TTLL5^MiEx/-^* and

*Figure 5 continued on next page*

*Figure 5 continued*

*TTLL5^E517A/-* were hemizygous over *Df(3R)BSC679*. The genotype for TTLL5^pBac/-^ was *MattubGal4/+; TTLL5^pBac^/Df(3R)BSC679*. The genotypes of the rescued animals were *MattubGal4/UASP-Venus::TTLL5; TTLL5 ^alleles^/Df(3R)BSC679*.

thus conclude that the onset of unidirectional fast streaming in S10B oocytes requires normal levels of *TTLL5*. Again, because abnormal streaming was observed when the enzymatic glutamylation activity of TTLL5 was impaired (*TTLL5^E517A/-^*), our results suggest that TTLL5 acts on the ooplasmic streaming through its role in MT glutamylation.

### TTLL5 affects kinesin distribution in the late-stage female germline

Both Staufen/*osk* mRNA localization and ooplasmic streaming depend on kinesin-1-driven processes (*Lu et al., 2016*; *Serbus et al., 2005*; *Brendza et al., 2000*). Therefore, we examined whether *TTLL5* somehow affects kinesin-1. Firstly, the expression levels of ovarian Kinesin heavy chain (Khc) were neither affected by lack of *TTLL5* function (*Figure 7A*) nor by *TTLL5* overexpression (*Figure 7B*). Next, we determined the localization of Khc in stage 10B, when the fast streaming starts. Even though the MT arrays start to disassemble and reorganize in the S10B oocyte, Khc showed a biased enrichment at the posterior in most wild-type control oocytes (*Figure 7C*). Out of 37 wild-type oocytes, 76% showed a posteriorly enriched Khc signal (*Figure 7D*). In contrast, Khc distribution showed a posterior oocyte enrichment in only 13% and 14%, respectively, in *TTLL5^pBac/-^* and *TTLL5^MiEx/-^* mutant egg chambers, respectively (*Figure 7C and D*). Elevated expression levels of TTLL5 seemed to affect Khc posterior accumulation as well because a smaller fraction of oocytes (54%) displayed clear posterior Khc enrichment than in the control (*Figure 7C and D*). These results demonstrate that normal levels of TTLL5 are needed for the proper posterior enrichment of Khc in S10B oocytes.

Khc showed an additional higher accumulation in the inner part of the oocytes (arrowhead in *Figure 7E*) and this was seen in 76% of the oocytes (*Figure 7F*). We measured the fluorescence intensity of the Khc signal along the anterior–posterior (AP) axis of the oocytes (line in *Figure 7E*). The signal intensity showed a peak in the center or more posteriorly in wild-type oocytes (*Figure 7G* and *Appendix 1—figure 4*), indicating that Khc concentration is higher and Khc might function in these regions of the oocyte. In the *TTLL5* mutants, the inner accumulation of Khc was strongly reduced or absent (*Figure 7E and G*, *Appendix 1—figure 4*). Only 19% of *TTLL5^pBac/-^* and 14% of *TTLL5^MiEx/-^* oocytes showed a slight inner Khc accumulation (*Figure 7F*). Overexpressing *TTLL5* in wild-type oocytes showed an inner peak of Khc accumulation in 65%, which is slightly less than the wild-type control. Besides, the center of the single peak appeared in a more anterior position (*Appendix 1—figure 4*). Interestingly, the higher levels of *TTLL5* also led to the appearance of two peaks of internal Khc accumulation in the oocyte, an anterior and a posterior one (*Figure 7E and G* and *Appendix 1—figure 4*). These peculiar inner accumulation patterns of Khc upon *TTLL5* overexpression might explain the unusual ooplasmic streaming patterns with a more anterior center and second, more posterior streaming center, that appeared in oocytes upon *TTLL5* overexpression. This suggests that Khc is

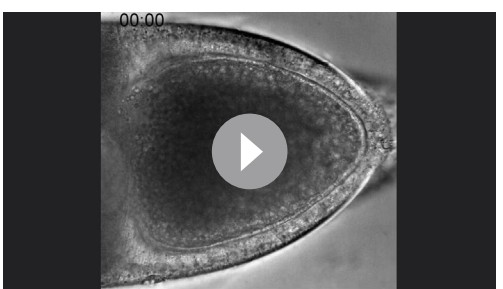

**Video 1.** Ooplasmic streaming in *w* (control): two oocytes.

https://elifesciences.org/articles/87125/figures#video1

**Video 2.** Ooplasmic streaming in *TTLL5^MiEx/-^*: two oocytes.

https://elifesciences.org/articles/87125/figures#video2

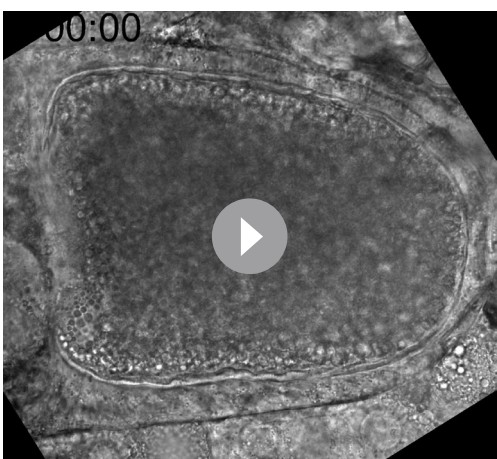

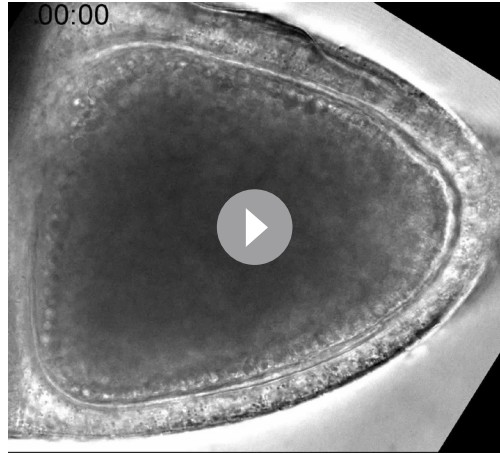

**Video 3.** Ooplasmic streaming in *TTLL5*^E517A/-^: two oocytes.
https://elifesciences.org/articles/87125/figures#video3

**Video 4.** Ooplasmic streaming in *Venus::TTLL5; TTLL5*^MiEx/-^ (rescued mutant): two oocytes.
https://elifesciences.org/articles/87125/figures#video4

causally connected to the fast cytoplasmic streaming, and both Khc localization and Khc-mediated streaming need *TTLL5*.

We also evaluated the localization of the Bicaudal-D (BicD) protein during the early stages of oogenesis. BicD is the linker that couples diverse cargos to the dynein/dynactin motor and can thus serve to evaluate the dynein transport (*Claussen and Suter, 2005*). The expression levels of BicD were not affected by the TTLL5 levels (*Appendix 1—figure 5*). Also, BicD localized to the posterior of the oocytes, and the ratio of the oocytes showing posteriorly localized BicD was similar in controls and *TTLL5* mutants (*Appendix 1—figure 5B and C*). In conclusion and as suggested by the normal appearance of the egg chambers (see next paragraph), at least the dynein/dynactin/BicD transport on early oogenesis microtubules does not appear to depend on *TTLL5*.

## Female fertility and ovarian development do not require *TTLL5*

The dramatic loss of polyglutamylated ovarian α-tubulin and the phenotypes during late oogenesis in *TTLL5* mutants incited us to analyze whether the morphology of the egg chambers from different stages and female fertility were affected in *TTLL5* mutants. However, the lack of *TTLL5* neither caused a significant decrease in hatching rates, nor distinguishable morphological changes in ovaries. The overall MT polarity of stage 10B oocytes was normal in *TTLL5* mutants based on the localization of the polarity markers Gurken and Staufen protein (*Appendix 1—figure 6*; *Neuman-Silberberg and Schüpbach, 1996*; *Zimyanin et al., 2007*).

## The glutamylation activity of *TTLL5* modulates the pausing of anterograde axonal transport of mitochondria

Previous studies in *Drosophila* reported that the absence of α-tubulin glutamylation did not seem to be detrimental to the nervous system functions tested in *Drosophila* (*Devambez et al., 2017*). The wing nerve is an additional, attractive system

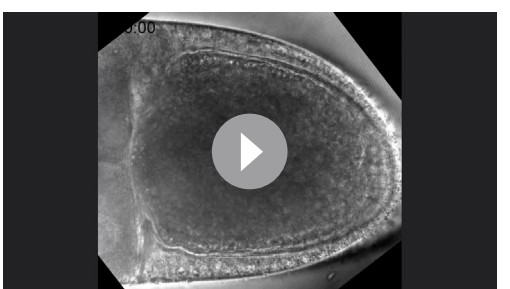

**Video 5.** Ooplasmic streaming in *Venus::TTLL5* (overexpressed): two oocytes.
https://elifesciences.org/articles/87125/figures#video5

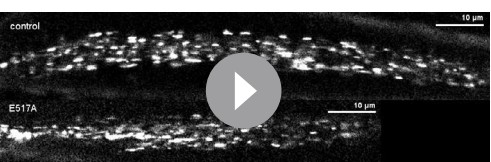

**Video 6.** Appl-Gal4>Mito::GFP transport in the L1 vein. Top movie: control; bottom movie: *TTLL5*^E517A/-^.
https://elifesciences.org/articles/87125/figures#video6

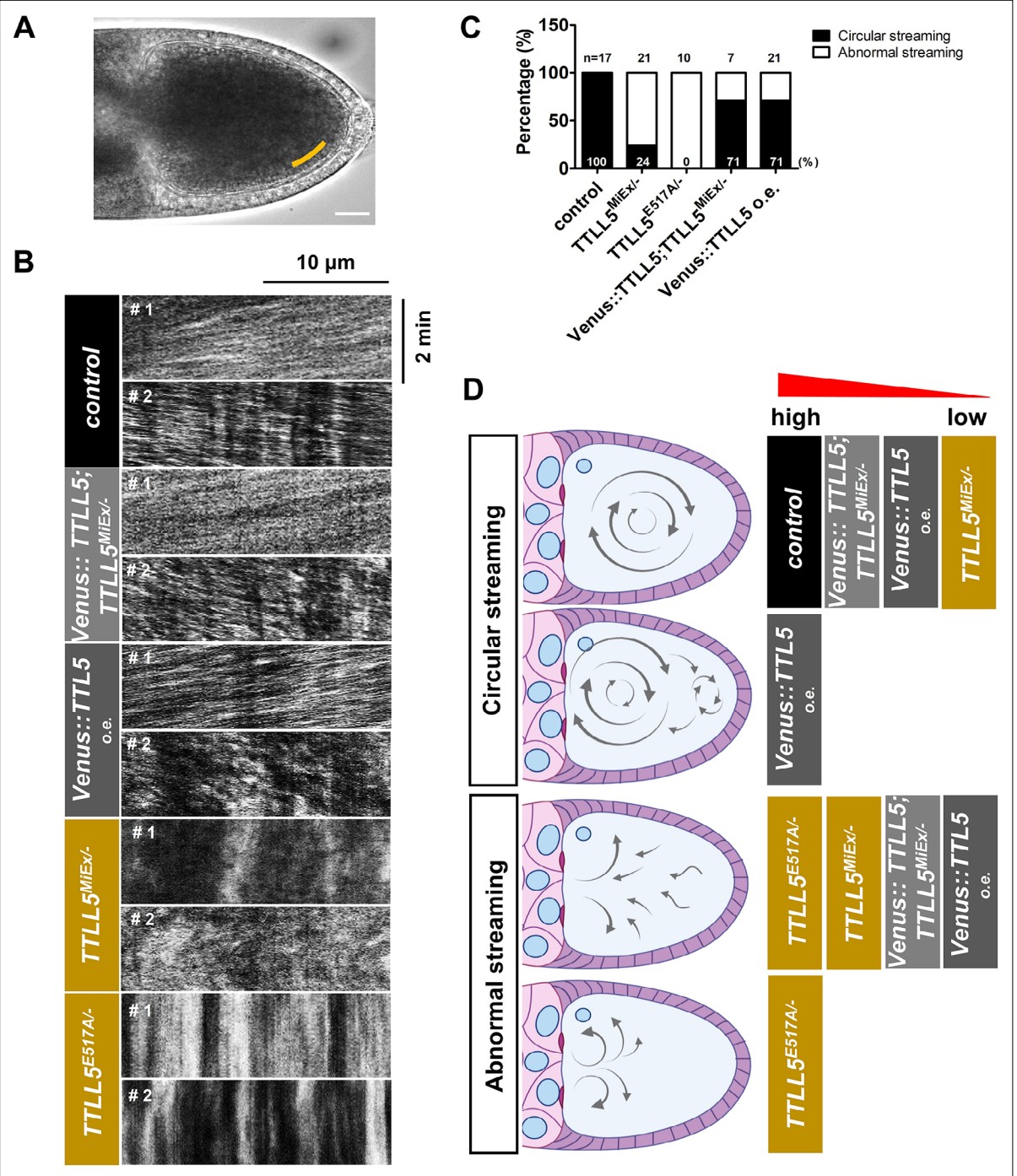

**Figure 6.** Role of *TTLL5* in fast ooplasmic streaming. (**A**) Example of an S10B oocyte used for particle flow measurements based on time-lapse movies of ooplasmic streaming (*Videos 1–5*). Scale bar: 25 μm. (**B**) Kymographs were generated along a line close to the posterior cortex on one side of the ooplasm (region marked by a yellow line in **A**). Sample kymographs were shown based on *Videos 1–5*. Streaming was shown from two representative oocytes for each movie. The first and second oocytes are marked as sample 1 (#1) and sample 2 (#2), respectively. The unidirectional streaming along the posterior cortex is seen in the two control and the *TTLL5* rescue samples, and in sample 1 of the *TTLL5* overexpressing oocyte. Disordered streaming is seen in the second sample of the *TTLL5* overexpressing line and both samples of the *TTLL5* mutants. (**C**) Quantification of streaming patterns seen in the different genotypes. 100% control (n = 17), 71% *TTLL5* rescued (n = 7), and 71% *TTLL5* overexpression (n = 21) S10B oocytes showed an overall circular unidirectional streaming pattern. In contrast, 76% *TTLL5^{MiEx/-}* (n = 21) and 100% *TTLL5^{E517A/-}* (n = 10) oocytes showed an abnormal streaming pattern. The stronger penetrance of the *TTLL5^{E517A/-}* phenotype (**C**) compared to the null mutant could be seen as evidence for a slight dominant effect of the point mutation. This is consistent with the stronger expressivity of the streaming defects observed in the movies (**B, D**), although the numbers appear a bit small to firmly conclude this. (**D**) Schematic representation of the different streaming patterns observed. The frequency with

*Figure 6 continued on next page*

Figure 6 continued

which each streaming pattern was observed in the different genotypes is indicated from high (left) to low (right). 'Circular streaming pattern' was further subdivided into 'central streaming pattern (top)' and 'anterior-biased streaming pattern (below).' The 'central streaming' was observed in all controls, 71% of the *TTLL5* rescue, 43% of the *TTLL5* overexpression, and 21% of the *TTLL5*^MiEx/- oocytes. The 'anterior-biased streaming' was frequently seen in 28% of the *TTLL5* overexpression ovaries, where the main circular center moved to the anterior part, leaving the posterior with a chaotic streaming flow. 'Abnormal streaming' included the oocytes that showed an overall chaotic flow direction (upper one) and the partially disrupted flow (below). The abnormal streaming patterns were mainly seen in the situations when *TTLL5* was insufficient or inactive, and at a low frequency also in *TTLL5* rescued and *TTLL5* overexpressing ovaries. The genotype of the control was *w* or *+/Df(3)BSC679*. The genotypes for *TTLL5*^MiEx/- and *TTLL5*^E517A/- were both over *Df(3)BSC679*. The genotype of the rescued flies was *MattubGal4/UAS-Venus::TTLL5; TTLL5*^MiEx/*Df(3)BSC679*. The genotype of *TTLL5* overexpressing flies was *MattubGal4/UAS-Venus::TTLL5*.

to monitor organelle transport, such as the movement of mitochondria (*Vagnoni and Bullock, 2016*; *Hollenbeck and Saxton, 2005*). To investigate the role of *TTLL5* and glutamylation for neuronal transport in the peripheral nervous system, we used wing nerves as a model system. Specifically, we tracked the movement of GFP-labeled mitochondria (Mito::GFP) within the axon bundles of the arch in the L1 region (*Vagnoni and Bullock, 2016*; *Figure 8A*). The mitochondria transported toward the thorax move by plus-end transport, which is driven by kinesin, whereas mitochondria transported oppositely are transported by dynein (*Vagnoni and Bullock, 2016*).

Time-lapse videos of GFP-labeled mitochondria show their movements in *Video 6*. Transport of mitochondria occurred mostly in a unidirectional manner during a 3 min tracing period. During the occasional bidirectional movement, mitochondria moved a short distance toward one direction before abruptly reversing their direction. A schematic unidirectional running pattern is shown in *Figure 8B*. The start of a run was chosen at the time point when a motile particle began to move away from a stationary position or when it entered the focal plane, and the stop of a run was set when the particle terminated moving or moved out of the focal plane in the given time window. The 'pausing' was called when the particle moved with less than 0.2 μm/s within the time window of a 'run.' Particles with speeds higher than 0.2 μm/s were counted as 'transported.' Motile mitochondria were manually tracked in both anterograde and retrograde directions. For the anterograde transport, the run length of mitochondria was similar between wild-type controls and *TTLL5*^E517A/- (*Figure 8C*). However, a significant increase was observed in the total run velocity in *TTLL5* ^E517A/- compared to wild-type controls (*Figure 8D*). Whereas the transport velocity of mitochondria was not affected (*Figure 8E*), the lack of the glutamylase activity had a striking effect on the pausing time of the anterograde transport of mitochondria, which was reduced by approximately 1.4 times (*Figure 8F*). This finding suggested that *TTLL5* may not directly influence the transport process but instead modulate the pausing events during mitochondrial transport.

To further study the pausing behavior of mitochondria, we generated kymographs to analyze the movement of individual motile mitochondria. Kymographs of three representative anterogradely transported mitochondria are shown for both genotypes (*Figure 8G*). The pausing events of the mitochondria are indicated by yellow arrowheads. Notably, a higher frequency of pausing was observed in the control group compared to the *TTLL5*^E517A/- mutant. By analyzing the pausing events within a given distance, we found that the control exhibited a 1.5 times greater frequency of pausing compared to the *TTLL5*^E517A/- mutant (*Figure 8H*). These results indicate that the longer total pausing time in controls can be attributed to a higher frequency of pausing events during anterograde transport.

In contrast, we did not observe significant changes in the parameters of retrograde transport in *TTLL5* mutants, indicating that dynein-based transport was not affected. To sum up, these results show that the glutamylation activity of TTLL5 plays a role in regulating the pausing of anterograde axonal transport. Although we have not provided direct evidence for the cell-autonomous nature of this modification, based on the more detailed analysis from the oocytes, likely, these effects are also cell-autonomous in the nervous system. In the next section, we will discuss recent results from other groups that support the existence of a detailed cell-autonomous pathway.

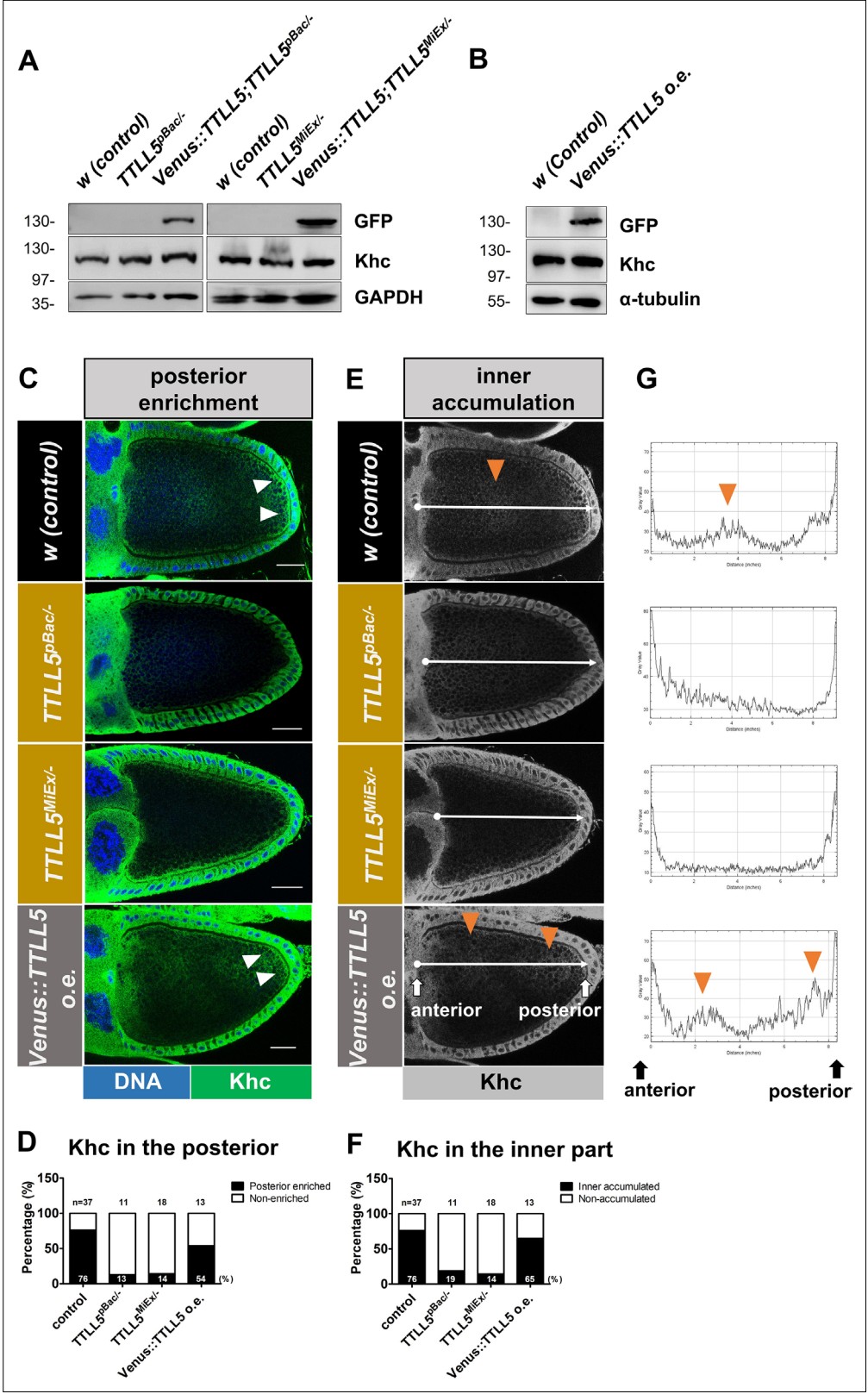

**Figure 7.** Role of *TTLL5* for kinesin distribution in ovaries. (**A**, **B**) Khc levels remained unchanged between *w* control, *TTLL5* deficient mutants, rescued *TTLL5* mutants, and *TTLL5* overexpressing ovaries. Anti-GFP antibodies reveal the expression of *Venus:TTLL5*. GAPDH served as the loading control. (**C**, **E**) Confocal micrographs showing the Khc distribution in stage 10B oocytes. (**C**) White arrowheads point to the posterior enrichment of Khc along the

*Figure 7 continued on next page*

*Figure 7 continued*

cortical region in the wild-type control and TTLL5 overexpressing oocytes. (**D**) Blind quantification of the posterior enrichment of Khc. The frequency of oocytes with posterior enrichment is shown graphically and numerically. N, number of oocytes evaluated. (**E**) A white line was drawn along the anterior–posterior axis of the oocyte using Fiji. The lines start at the nurse cell oocyte border on the left and end at the posterior follicle cells. The line width was 50 units. (**G**) Intensity charts were plotted based on the line drawn along the AP axis in (**E**). The orange arrowheads in the oocytes and the intensity charts point to the regions showing higher Khc accumulation in the inner region of the oocyte. (**F**) Fraction of the oocytes that showed inner hyperaccumulation of Khc were quantified. N, number of oocytes evaluated. The percentage of oocytes showing inner enrichment of Khc is shown graphically and numerically. Judging Khc distribution was done blindly by two persons. The fractions were taken from the mean values obtained from the two persons. Scale bar: 25 µm. The Khc signal is shown in green, and Hoechst in blue. The genotype for controls was *w* or *+/Df(3R)BSC679*. Genotype for *TTLL5^pBac/-^: MattubGal4/+, TTLL5^pBac^/Df(3R) BSC679; TTLL5^MiEx/-^: TTLL5^MiEx^/Df(3R)BSC679*. Genotype for *TTLL5* overexpression: *MattubGal4/UAS-Venus::TTLL5*.

The online version of this article includes the following source data for figure 7:

**Source data 1.** Source data for *Figure 7*.

## Discussion

### Glutamylation of the C-terminal tail of ovarian α-tubulin

This article describes the sidechain glutamylation of the C-terminal domain of α-tubulin in *Drosophila* ovaries as identified with a combination of using specific antibodies and MS analyses. Results from wild-type ovaries and mutants revealed that *TTLL5* is essential for these modifications. Consistent with results from mammalian cells and the *Drosophila* nervous system (*Devambez et al., 2017*; *van Dijk et al., 2007*), our combined western blot and MS results show a requirement for TTLL5 for the accumulation of α-tubulin with the first branching Glu sidechain added to one of the C-terminal Glus. In theory, this could mean that *TTLL5* is required for the addition of this first Glu sidechain residue or it might be required for the stability of glutamylated α-tubulin. TTLL5 has the enzymatic activity to add glutamyl groups and mammalian TTLL5 is considered to be an initiator glutamylase (*van Dijk et al., 2007*). We can, therefore, assume that the former is the case in *Drosophila*, too. TTLL1 and TTLL6 are elongators of polyglutamylation of α-tubulin (*Gadadhar et al., 2017*; *van Dijk et al., 2007*; *Janke et al., 2005*). However, there is no evidence that the homologous *TTLL6A/TTLL6B* or *TTLL1A/1B* are expressed in *Drosophila* ovaries (*Supplementary file 1*). The robust reduction of polyglutamylated tubulin in *TTLL5* mutants observed in ovaries (this work) and the nervous system (*Devambez et al., 2017*) further shows that *TTLL5* is also needed, directly or indirectly, for polyglutamylation in *Drosophila*. However, we cannot easily figure out in vivo if TTLL5 itself has the activity to extend the oligo-Glu sidechains.

The MS results revealed that glutamylation strongly preferred the substrate αTub84B/D over αTub67C, even though αTub67C is an ovarian-specific α-tubulin. This points to a possible need for a specific α-tubulin variant in the germline that is resistant to sidechain modifications. This resistance may serve to regulate different transport processes by providing local MT sites with reduced sidechain glutamylation, which allows for longer processive movement of kinesin motors as was observed in the absence of glutamylation in the mitochondrial transport assay (*Figure 8*). It would be interesting to find out whether MT tracks containing specific hypoglutamylated regions exist in oocytes and even more, whether such regions serve to more effectively transport and stream cytoplasmic components over long distances in large cells like the oocyte.

### Function of glutamylation of microtubules on kinesin-1-dependent streaming in late-stage oocytes

The unidirectional flow seen during rapid ooplasmic streaming is caused by the kinesin-1-dependent MT sliding and MT cargo transport along the cortical and subcortical MTs (*Lu et al., 2016*; *Monteith et al., 2016*). The proper localization of Khc to the posterior cortical region (*Figure 7*) and the glutamylation of cortical MTs (*Figure 4*) depend on TTLL5, which is normally also concentrated at the cortex in S10B oocytes (*Figure 5C* and *Appendix 1—figure 3*). All these observations point to the role of glutamylation of the MTs as a precondition for the proper function of kinesin-1.

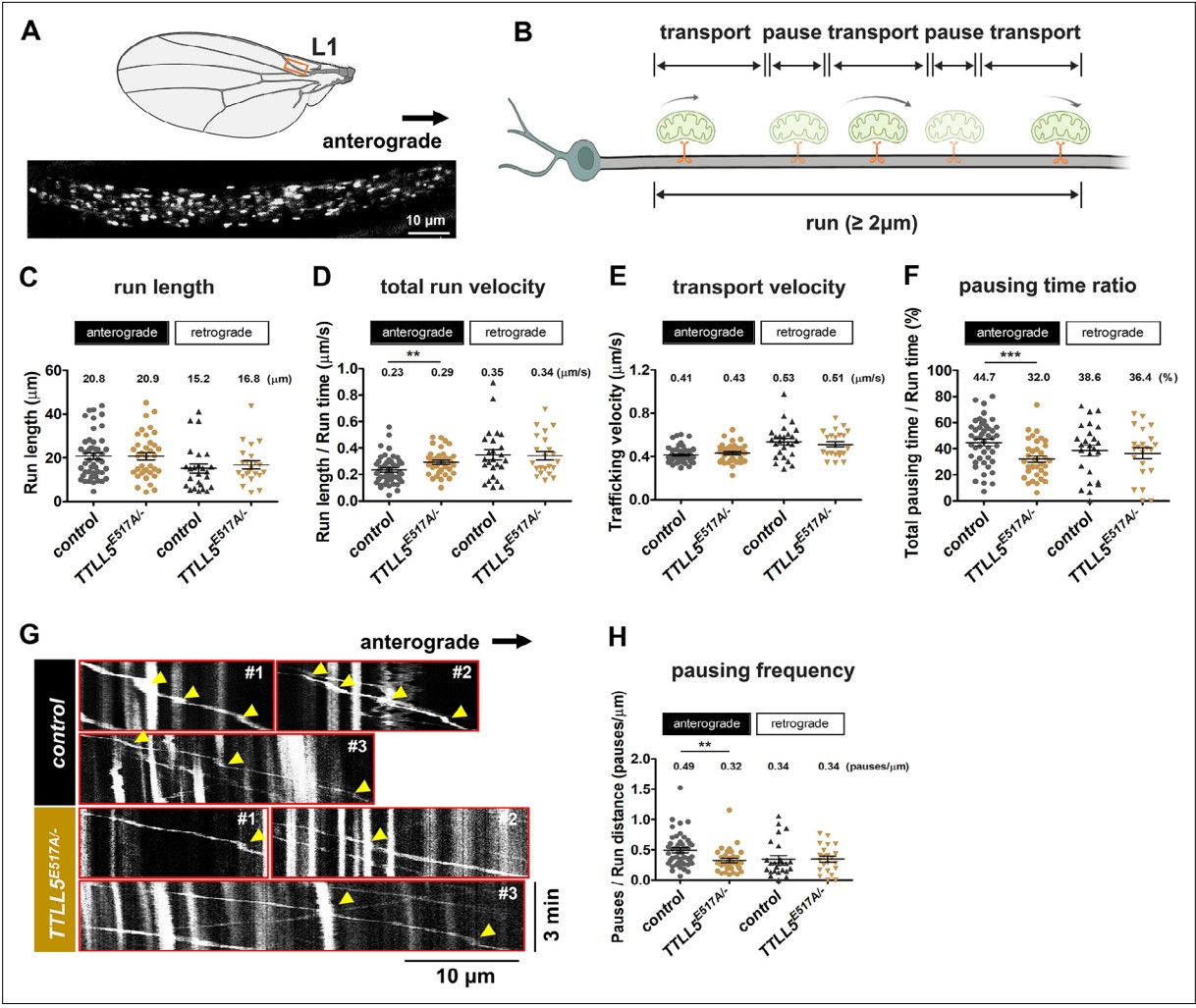

**Figure 8.** Role of *TTLL5* in the transport of mitochondria in the L1 wing nerve. (**A**) Representative still image of GFP-labeled mitochondria in axons of the L1 vein of both control and *TTLL5*^E517A/-^ wings. The position of the L1 region is indicated in the picture above. (**B**) Scheme of the transport behavior of the mitochondria in the wing neuron. See the main text for a detailed description. The schema was drawn by biorender.com. (**C–F**) Mean values of run length, total run velocity, transport velocity, and pausing time ratio in the control and the *TTLL5* ^E517A/-^ mutant for both anterograde and retrograde transport. (**C**) The run length is the sum of the distances traveled by individual mitochondria in a transporting state. (**D**) The total run velocity indicates the run distance divided by the run time. The run time is the sum of the total pausing time plus the total transporting time. (**E**) Transport velocity indicates the mean of the instant velocity of the transported mitochondria. (**F**) The pausing time ratio was calculated as total pausing time divided by run time. The total pausing time was the sum of the time when the mitochondria were pausing. (**G**) Kymographs were generated from three representative mitochondria in the control and the *TTLL5*^E517A/-^ mutant. Yellow arrowheads indicate the pauses of selected mitochondria over the 3 min time window. (**H**) The mean values of pausing frequency. Pausing frequency indicates the number of pauses divided by run length. Mitochondria were quantified from seven control fly wings and 8 *TTLL5*^E517A/-^ fly wings. The numbers of mitochondria (n) analyzed for the anterograde control, anterograde *TTLL5*^E517A/-^, retrograde control, and retrograde *TTLL5*^E517A/-^ were 46, 36, 25, and 21, respectively. ***$p<0.005$, **$p<0.01$. Genotypes for controls were a mixture of *ApplGal4;UAS-Mito::GFP;+/Df(3R)BSC679* and *ApplGal4;UAS-Mito::GFP;TTLL5* ^E517A/+^. Genotypes for *TTLL5*^E517A/-^ mutants were *ApplGal4; UAS-Mito::GFP; TTLL5*^E517A^/*Df(3R)BSC679*.

Kinesin-1 also accumulates at higher levels in the inner part of the oocyte (*Figure 7*), suggesting that the inner-oocyte MTs might also contribute to the rapid streaming observed. This hypothesis is supported by the observation that hyperglutamylation by TTLL5 overexpression often leads to an anterior-biased accumulation of Khc in the interior of the oocyte or the formation of two inner Khc clusters along the anterior–posterior axis (*Figure 7E and G*, *Appendix 1—figure 4*). This abnormal Khc distribution is then mirrored by the streaming pattern in such oocytes (*Figure 6D*), suggesting that it may cause the shift of the streaming center to the anterior part of the oocytes and the appearance of a second posterior streaming center. Further investigations through direct tracking of Khc

movement in S10B oocytes under hypo- and hyperglutamylation conditions may provide more insight into the impact of MT glutamylation on kinesin-1 function in vivo.

## Role of glutamylation of α-tubulin for stopping transported mitochondria

The previous study showed that α-tubulin was the only tubulin subunit that can be glutamylated in the *Drosophila* nervous system and *TTLL5* is required for α-tubulin glutamylation in the neurons (*Devambez et al., 2017*). However, the absence of *TTLL5* did not reveal clear defects in larval neuromuscular junction morphology, larval locomotion behavior, or vesicular axonal transport in larval neurons. We, therefore, used a different system, the transport of mitochondria in the adult wing axon, to test for a possible neuronal function.

By measuring the parameters related to the pausing time and pause frequency, we found that glutamylated α-tubulins promoted the normal stopping or pausing of mitochondria during axonal transport. Glutamylated sidechains of α-tubulin might act as marks or even obstacles that disturb the movement of mitochondria. Consistent with our results, the previous *Drosophila TTLL5* study noted a slight effect on the pausing of vesicles during axonal transport in *TTLL5* mutants even though this turned out to be not significant (*Devambez et al., 2017*). Vesicle transport in larval neurons may exhibit a lower sensitivity to MT glutamylation than mitochondria transport in adult wing neurons. Also, *Zheng et al., 2022* reported that different cargo could have different modification requirements for MT transport. The basis for these differences could be studied systematically in future experiments in the wing vein system. Moreover, a previous study revealed a similar function of glutamylation of α-tubulin for the transport of mitochondria (*Bodakuntla et al., 2021*). The absence of polyglutamylation of α-tubulin in *TTLL1*$^{-/-}$ increased the motility of mitochondria in the mouse neurons but did not affect their transport velocity. The results led to their hypothesis that glutamylation of α-tubulin may not directly affect transport but rather affect mitochondrial docking and motor binding to MT tracks by acting as roadblocks on the MT tracks.

Our results showed a selective effect of Glu-MTs on mitochondrial motility for plus-end transport in axons, a process that is carried out by kinesin-1 (*Pilling et al., 2006*). This suggests that glutamylation of MTs plays a role in controlling kinesin-1 activity similar to what we had observed in the stage 10B oocyte. Several in vitro studies have also shown that Glu-MTs affect kinesin-1 activity. The considerable negative charge of the modified regions in the poly-Glu-modified tubulins might interfere with the interaction of kinesin with tubulin (*Figure 1B*; *Skiniotis et al., 2004*). In vitro studies demonstrated that the artificial tethering of glutamyl peptides to the sidechains of MTs affected the function of kinesin-1 (*Sirajuddin et al., 2014*). More recent in vitro studies further reveal that TTLL7-modified β-tubulin polyglutamylation specifically decreases the motility of kinesin-1 (*Genova et al., 2023*). Moreover, the in vivo study by *Lu et al., 2016* has identified the MT binding site in the coiled coil region of Khc that is specifically required for the MT sliding in S10B oocytes. All these observations suggest that sidechain glutamylation of MTs is sensed by kinesin-1 and functions to interfere with the MT binding of kinesin-1, thus disturbing the movement of kinesin-1-dependent cargos.

Directly tracking kinesin-1 motor transport using the adult wing axon system could provide valuable insights. The *Drosophila* nervous system is known to have α-tubulin as the major substrate for glutamylation (*Devambez et al., 2017*), making it a powerful and simple in vivo model to study the effect of glutamylation of specific α-tubulin isotypes and their impact on MT motor movement. Additionally, the wing vein system presents an opportunity to examine the effects of various microtubule-associated proteins (MAPs), MT severing proteins, and other candidates on this transport in both wild-type and *TTLL5* mutants in vivo. To fully understand the mechanisms underlying the effect of glutamylated tubulin tails on MT transport, additional in vitro studies will be necessary. Such studies with defined components will help to determine whether glutamylated tubulin tails directly affect the motor itself or have effects on the MT transport through MAPs, modification enzymes, or other processes.

Even though an essential function of α-tubulin glutamylation has not emerged from this study, it has become clear that kinesin-1-dependent processes are modified or controlled by this modification and that evolution has adapted the tubulin code to optimize the usage of MTs for different evolving needs. That *TTLL5* is not essential for survival also makes this protein an interesting candidate drug

target to modulate its glutamylation activity to regulate and possibly even increase MT transport in neurodegenerative diseases where MT transport is known to be affected (*Bodakuntla et al., 2020*).

## Materials and methods

### *Drosophila* genetics

The fly stocks used in this study, *MattubGal4* (7062), *TTLL5^pBac^* (16140; allele name *TTLL5^B093^*), *TTLL5^Mi^* (32800; allele name *TTLL5^MI01917^*), and *Df(3R)BSC679* (26531), were obtained from the Bloomington Drosophila Stock Center. The *TTLL5^MiEx^* mutation was made by imprecise excision of the Minos element, which left a 3-base insertion (CTA), which introduced a premature stop codon at the genomic position 8132 in the open-reading frame (polypeptide position 392). The different *TTLL5* point mutations were generated by CRISPR/Cas9. The two *Drosophila* stocks used for *TTLL5* gene editing, *y[1] M{w[+mC]=nos-Cas9.P}ZH-2A w[\*]* and *y[1] v[1] P{y[+t7.7]=nos-phiC31\int.NLS}; P{y[+t7.7]=CaryP} attP40,* were obtained from the Bloomington Drosophila Stock Center (54591) (*Port et al., 2014*) and (25709), respectively. Target sequences near the K282, R339, and E517 codons were selected to introduce INDELs and the following point mutations: R339A (b6-7 loop), E517A, and P522A. Phosphorylated and annealed primers (*Supplementary file 1*) needed to construct the corresponding coding sequences for a single-guide RNA (sgRNA) were each cloned into the BbsI site of *pCFD5* according to the protocol published on http://www.crisprflydesign.org/. After amplifying the vector in XL1 blue cells, the plasmids were sequenced to verify the correct assembly. Subsequently, they were injected into *attP40* embryos. *TTLL5*-guide stocks with integrated genes encoding sgRNAs were established (=*v; attP40{v⁺, TTLL5-guide}/CyO*). To obtain INDELs for TTLL5, a three-generation crossing schema was set up. Firstly, *nos-cas9* virgins were crossed with males from the different TTLL5-guide stocks. Secondly, the subsequent single male progeny was crossed with *w; Ly/TM3, Sb* virgins. Thirdly, resulting in single *TM3, Sb* males were again crossed with *w; Ly/TM3, Sb* virgins to establish the stocks. The nature of the mutated *TTLL5* alleles was determined by sequencing the targeted region in each stock and comparing it to the sequence of the *TTLL5* gene of the parental *y w {w⁺, nos-cas9}* mothers and *TTLL5-guide* fathers (*Supplementary file 1*). All the *TTLL5* alleles were crossed to *Df(3R) BSC679* to analyze the effect of the *TTLL5* mutations in hemizygous animals. Abnormally shaped egg chambers were seen in *TTLL5^pBac^/Df(3R)BSC679*. But it seems to be the second site effect as the phenotype disappeared when it was in combination with specific second chromosomes, for example, *MattubGal4/+ or +/UAS-Venus::TTLL5.*

The *UASp-venus::TTLL5* DNA construct was made from the *TTLL5* cDNA amplified by PCR (forward primer: 5'-GTTCAGATCTATGCCTTCTTCATTGTGTG-3'; reverse primer: 5'-GAGTCATTCTAGAGCTTCATAGAAATACCTTCTCC-3'). The resulting DNA was inserted into the *pUASP-venus* vector via Xba I/Bgl II ligation and targeted into the *attP-58A* landing platform (24484) to generate the transgenetic *UASp-venus::TTLL5* fly strain. *ApplGal4* and *UAS-mito::GFP* were kindly provided by Simon Bullock's group (*Vagnoni and Bullock, 2016*).

### Western blotting

Western blotting performed with ovarian extracts was carried out with 10–20 pairs of ovaries for each genotype. Newly hatched females of the indicated genotypes were collected and crossed with *w* males in fresh vials with ample yeast food for 2–3 d. Fully developed ovaries were then dissected and collected. The samples were directly lysed in Laemmli buffer, boiled at 95°C for 10 min, fractionated by SDS-PAGE, and transferred onto PVDF membranes. The following primary antibodies were used: mouse 1D5 (1:500, Synaptic System), mouse GT335 (1:200, Adipogen), mouse DM1A (1:1000, Sigma), mouse tyrosinated α-tubulin (1:1000, Sigma), rabbit GFP (1:2000, Sigma), rabbit GAPDH (1:1000, GeneTex), rabbit *Drosophila* Clathrin light chain (Clc) (1:2000, *Heerssen et al., 2008*), mouse BicD (1B11 plus 4C2, 1:5) (*Suter and Steward, 1991*), rabbit Khc (1:1000, cytoskeleton), and mouse TAP952 (1:500, Merck). For testing the expression of TTLL5, the ovaries were homogenized in ice-cold lysis buffer (150 mM NaCl, 50 mM Tris-HCl [pH 7.5], 1 mM MgCl₂, 1 mM EDTA, 0.1% Triton X-100, 0.5 mM DTT, and protease inhibitor cocktail [Roche]) and centrifuged at 13,200 × *g* for 20 min. The supernatant with the soluble proteins was collected and loaded for western blotting. The rabbit anti-Drosophila TTLL5 (1:500) was generated by the GenScript company. The fragment used for antigen induction was: TKLLRKLFNVHGLTEVQGENNNFNLLWTGVHMKLDIVRNLAPYQRVNHFPRSYEMTRKDR

LYKNIERMQHLRGMKHFDIVPQTFVLPIESRDLVVAHNKHRGPWIVKPAASSRGRGIFIVNSPDQIPQDEQA
VVSKYIVDPLCIDGHKCDLRVYVLVTSFDPLIIYLYEEGIVRLATVKYDRHADNLWNPCMHLCNYSINKYHS
DYIRSSDAQDEDVGHKWTLSALLRHLKLQSCDTRQLMLNIEDLIIKAVLACAQSIISACRMFVPNGNNCFEL
YGFDILIDNALKPWLLEINLSPSMGVDSPLDTKVKSCLMADLLTCVGIPAYS. Secondary antibodies were
HRP-linked goat anti-mouse (1:10,000, GE Healthcare) and HRP-linked goat anti-rabbit (1:10,000, GE
Healthcare). Antibodies were incubated in 5% non-fat milk PBST (PBS with 0.1% Triton X-100) and
washed with PBST. Chemiluminescence detection was performed with the Amersham imager 600 (GE
Healthcare). Western blotting results were replicated at least three times. Most are biological repli-
cates, but some are technical replicates.

## Mass spectrometry

Proteins were extracted from the ovaries of 100 three-day-old, well-fed females of each genotype,
crossed to wild-type males, and proteins were boiled and run on SDS-PAGE. The gel was then stained
with Coomassie blue. The gel pieces containing the α-tubulin region were cut, alkylated, and digested
by trypsin N (ProtiFi) for 3 hr at 55°C. C-terminal peptides of α-tubulin were analyzed by liquid chroma-
tography (LC)-MS/MS (PROXEON coupled to a QExactive mass spectrometer, Thermo Fisher Scien-
tific). Samples were searched with TPP (Trans-Proteomics Pipeline) against the uniprotKB *Drosophila*
database (December 2017). PEAKS, EasyProt, and MSfragger were also used as search engines to
identify the modifications. Peptides were identified by PSM value (peptide spectrum match). To avoid
repetition of total PSM quantification, we searched only for one type of PTM in an analysis each time.
Total PSM indicates the sum of PSMs of primary C-terminal peptides with either glutamylation or no
modification. Data are available via ProteomeXchange with identifier PXD035270. 'Data are available
via ProteomeXchange with identifier PXD035270.' Reviewer account details: reviewer_pxd035270@
ebi.ac.uk Password: mmvkUOWK.

## Ooplasmic streaming in living oocytes

Three-day-old females were kept together with wild-type males under non-crowding conditions at
25°C on food containing dry yeast. The determination of ooplasmic streaming was based on the
method of *Lu et al., 2016*. Ovaries were directly dissected in 10S halocarbon oil (VWR), and egg
chambers were teased apart in the oil drop on 24 × 50 mm coverslips (VWR). Living Stage 10B oocytes
were directly observed and imaged under the 60 ×1.4 NA oil immersion objective on a Nikon W1 LIPSI
spinning disk with a 2× Photometrics Prime BSI CMOS camera. Time-lapse images were acquired
every 2 s for 2 min under bright-field (BF) controlled by Nikon Elements software. Representative
movies were selected from at least three independent experiments and n is described in the figure
legend. Flow tracks were generated by kymographs using Fiji (*Schindelin et al., 2012*) plugin Multi
Kymograph (https://imagej.net/Multi_Kymograph).

## Immunostaining and confocal microscopy

Ovaries were dissected in 1× PBS and fixed in buffer (200 µl 4% formaldehyde [Sigma], 600 ul Heptane
[Merck Millipore], 20 ul DMSO [Sigma] for each group) for 20 min. Ovaries were then blocked in
2% milk powder in PBST (PBS, 0.5% Triton X-100). Samples were incubated with rabbit anti-Staufen
(1:2000) (*St Johnston et al., 1991*), mouse anti-Gurken (1D12, 1:10) (*Queenan et al., 1999*), mouse
anti-BicD (1B11 plus 4C2, 1:5) (*Suter and Steward, 1991*), rabbit anti-Khc (1:150, Cytoskeleton),
rabbit anti-GFP (1:200, Sigma), and mouse anti-1D5 (1:150, Synaptic System) at 4°C overnight, and
incubated with secondary Alexa Fluor goat anti-mouse Cy3 (1:400, Jackson ImmuonoResearch) and
Alexa Fluor goat anti-rabbit 488 (1:400, Life Technologies, Thermo Fisher) at room temperature for
3 hr. DNA and nuclei were stained using Hoechst 33528 (Thermo Fisher Scientific) for 10 min. Mounted
samples were imaged with a Leica TCS SP8 confocal microscope equipped with a 63× 1.40 oil lens
and controlled by the Las X software (Leica). Images of single focal planes and 12 µm Z projection
were acquired. The crescent length of Staufen was processed by the Fuji segmented line tool based
on a single plane. The data were shown as means ± SEM by Prism 5 (GraphPad Software). The statis-
tical significance was determined by an unpaired two-tailed non-parametric *t*-test (Mann–Whitney
test). For the Khc intensity analysis, the intensity of fluorescence in the oocyte was plotted and the
intensity charts were generated using Fiji's 'Analyze-Plot profile' function. Representative images were
selected from three independent experiments.

### Mitochondrial transport in *Drosophila* wing neurons

Mitochondrial transport in *Drosophila* wing neurons was tracked as described (*Vagnoni and Bullock, 2016*). In brief, 1–2-day-old flies were anesthetized with $CO_2$ and immobilized in a custom-built chamber formed of two No. 1.5 24 ×50 mm coverslips (VWR). Imaging of wings was performed with a Nikon W1 LIPSI spinning disk microscope equipped with a 2× Photometrics Prime BSI CMOS camera and a 60 ×1.4 NA oil immersion lens. Time-lapse video of a single focal plane was acquired with an acquisition rate of 0.5 frame/s for 3 min for Mito::GFP using a 488 nm laser.

Quantification of the movements was performed in Fiji. At least seven wings of each genotype from two independent experiments were tracked for quantification. A region of the wing arch of the L1 vein was selected and straightened with the Straighten plugin (Eva Kocsis, NIH, MD). For each 'run,' the total transport distance of more than 2 μm was defined as 'motile mitochondria.' To exclude the oscillatory movements of mitochondria, a minimal instant velocity between each frame was defined at 0.2 μm/s. Transported mitochondria were manually tracked by the manual tracking plugin (Fabrice Cordelières, Institute Curie, France). Kymographs of individual mitochondria were produced using Fiji (*Schindelin et al., 2012*) plugin Multi Kymograph (https://imagej.net/Multi_Kymograph). Run length, instant velocity values, and total running, transport and pausing times were exported into Excel for analysis. The data are shown as means ± SEM by Prism 5 (GraphPad). The statistical significance was determined with the Student's *t*-test.

### Sequence alignment

Sequences were aligned by CLC Sequence Viewer 8.0 software (QIAGEN). The UniProtKB accession numbers are as follows: *HsTTL*: Q8NG68; *HsTTLL5*: Q6EMB2; *MmTTLL5*: Q8CHB8; *Dm*TTLL5: Q9VBX5.

## Acknowledgements

We thank Simon L Bullock (Cambridge, UK) for *ApplGal4* and *UAS-mito::GFP* fly lines, Michel Steinmetz (Paul Scherrer Institute, Switzerland), and the Carsten Janke (Institute Curie, France) laboratory members for discussions and suggestions. We also thank the University of Bern (Switzerland) Mass Spectrometry Center members Sophie Braga Lagache and Natasha Buchs for performing the MS experiments and Manfred Heller and Anne-Christine Uldry for peptide analysis and discussion of the results. This work was financially supported by the project grants 31003A_173188 and 310030_205075 from the Swiss National Science Foundation (SNF; https://www.snf.ch/en) to BS. Equipment support was by an equipment grant from SNF (316030_150824) to BS and by the University of Bern (https://www.unibe.ch). Additional salary and material support were received from the University of Bern (https://www.unibe.ch) to BS. The funders had no role in the study design, data collection, analysis, decision to publish, or manuscript preparation.

## Additional information

### Funding

| Funder | Grant reference number | Author |
| --- | --- | --- |
| Swiss National Science Foundation | project grant 31003A_173188 | Beat Suter |
| Swiss National Science Foundation | Project Grant 310030_205075 | Beat Suter |
| University of Bern | | Beat Suter |

The funders had no role in study design, data collection and interpretation, or the decision to submit the work for publication.

### Author contributions

Mengjing Bao, Conceptualization, Data curation, Formal analysis, Validation, Investigation, Visualization, Writing – original draft, Writing – review and editing; Ruth E Dörig, Dirk Beuchle, Resources,

Investigation, Methodology; Paula Maria Vazquez-Pianzola, Formal analysis, Validation, Methodology; Beat Suter, Conceptualization, Formal analysis, Funding acquisition, Project administration, Writing – review and editing

**Author ORCIDs**
Beat Suter ![ORCID] http://orcid.org/0000-0002-0510-746X

**Decision letter and Author response**
Decision letter https://doi.org/10.7554/eLife.87125.sa1
Author response https://doi.org/10.7554/eLife.87125.sa2

## Additional files

### Supplementary files
• MDAR checklist

• Supplementary file 1. Tissue specific expression of the 11 *Drosophila TTLL* genes. (**a**) Tissue--specific expression of the 11 *Drosophila* TTLL genes. (**b**) Peptide sequence and frequency of unmodified full primary C-terminal sequence of α-tubulins identified by MS. (**c**) The full primary C-terminal sequence of α-tubulins modified with E sidechain modifications. The probabilities of E sidechain modifications in each peptide according to TPP are indicated for the four positions close to the C-term. (**d**) The full primary C-terminal sequence of α-tubulins modified by E sidechain modifications. The probabilities of E sidechain modifications in each peptide according to EasyProt, MSfragger, and PEAKS are indicated for the four positions close to the C-term. (**e**) Primers for cloning the sequences encoding the sgRNAs into the Drosophila transformation vector pCFD5. (**f**) Templates to introduce point mutations.

### Data availability
MS analyses produced datasets. These data were deposited and can be accessed with the following accession: PXD035270.

The following dataset was generated:

| Author(s) | Year | Dataset title | Dataset URL | Database and Identifier |
|---|---|---|---|---|
| Bao M, Dörig R, Vazquez-Pianzola P, Beuchle D, Suter B | 2023 | Differential modifications of the C-terminal tails of alpha-tubulin isoforms and their importance for kinesin-based microtubule transport in vivo | https://www.ebi.ac.uk/pride/archive/projects/PXD035270 | PRIDE, PXD035270 |

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

## Appendix 1

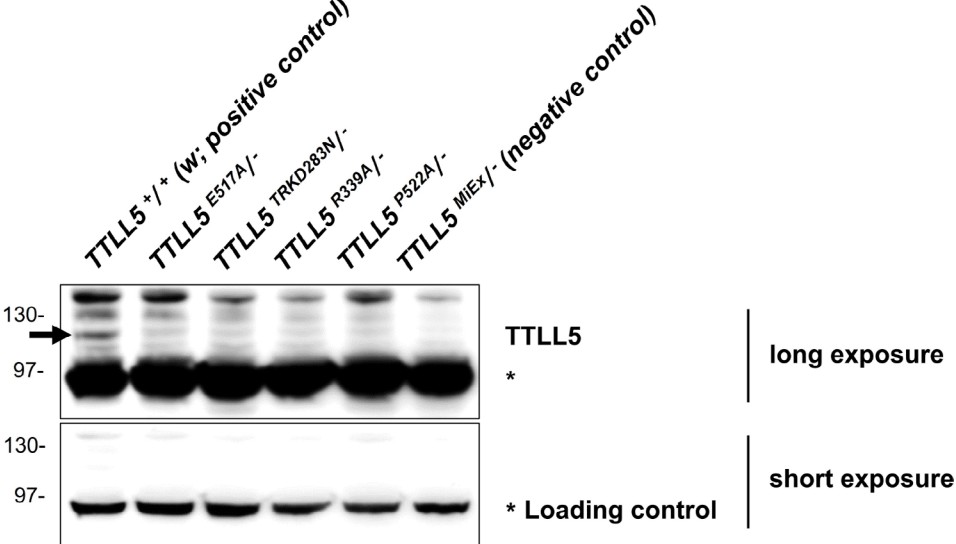

**Appendix 1—figure 1.** TTLL5 was stably expressed in all Crispr/Cas9 generated *TTLL5* mutants. All Crispr/Cas9 generated *TTLL5* mutations were analyzed hemizygously over *Df(3R)BSC679*. TR**K**D283N refers to a more complex mutation in which the four codons TR**K**D283 were replaced by a single N. We expect extracts from these animals to contain at most half the normal amount of TTLL5. *w* (*TTLL5⁺/TTLL5⁺*) was the positive control and the null mutation *TTLL5^MiEx^/Df(3R)BSC679* was the negative control. The signal of an unspecific band (labeled with *) under short exposure was treated as a loading control. *w*, which contains two copies of the *TTLL5⁺* allele expressed the highest levels of TTLL5. Crispr/Cas9 generated *TTLL5* mutants with a single allele of *TTLL5** expressed less TTLL5 compared to *w*, but more compared to the null mutant *TTLL5^MiEx^/Df(3R)BSC679*. The TTLL5 levels of the hemizygous point mutants were similar to the ones of the hemizygous *TTLL5⁺* (see **Figure 2B**). The faint band seen in *TTLL5^MiEx^/Df(3R)BSC679* is probably a background band.

The online version of this article includes the following source data for appendix 1—figure 1:

**Appendix 1—figure 1—source data 1.** Source data for *Appendix 1—figure 1*.

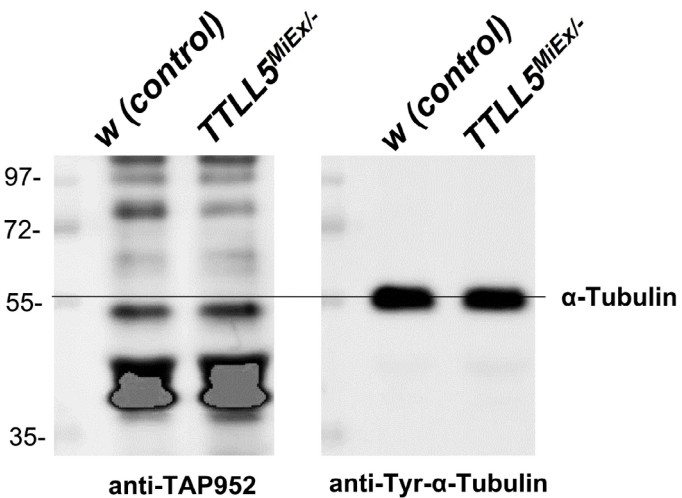

**Appendix 1—figure 2.** Glycylation of α-tubulin was not detected by western blotting from ovarian extracts. The glycylation of α-tubulin was evaluated by western blotting with the anti-monoglycylated Tubulin antibody TAP952. The anti-Tyr-α-Tubulin was a loading and size control. TAP952 did not detect a clear band in the α-Tubulin region in the wild type and *TTLL5* mutant.

The online version of this article includes the following source data for appendix 1—figure 2:

**Appendix 1—figure 2—source data 1.** Source data for *Appendix 1—figure 2*.

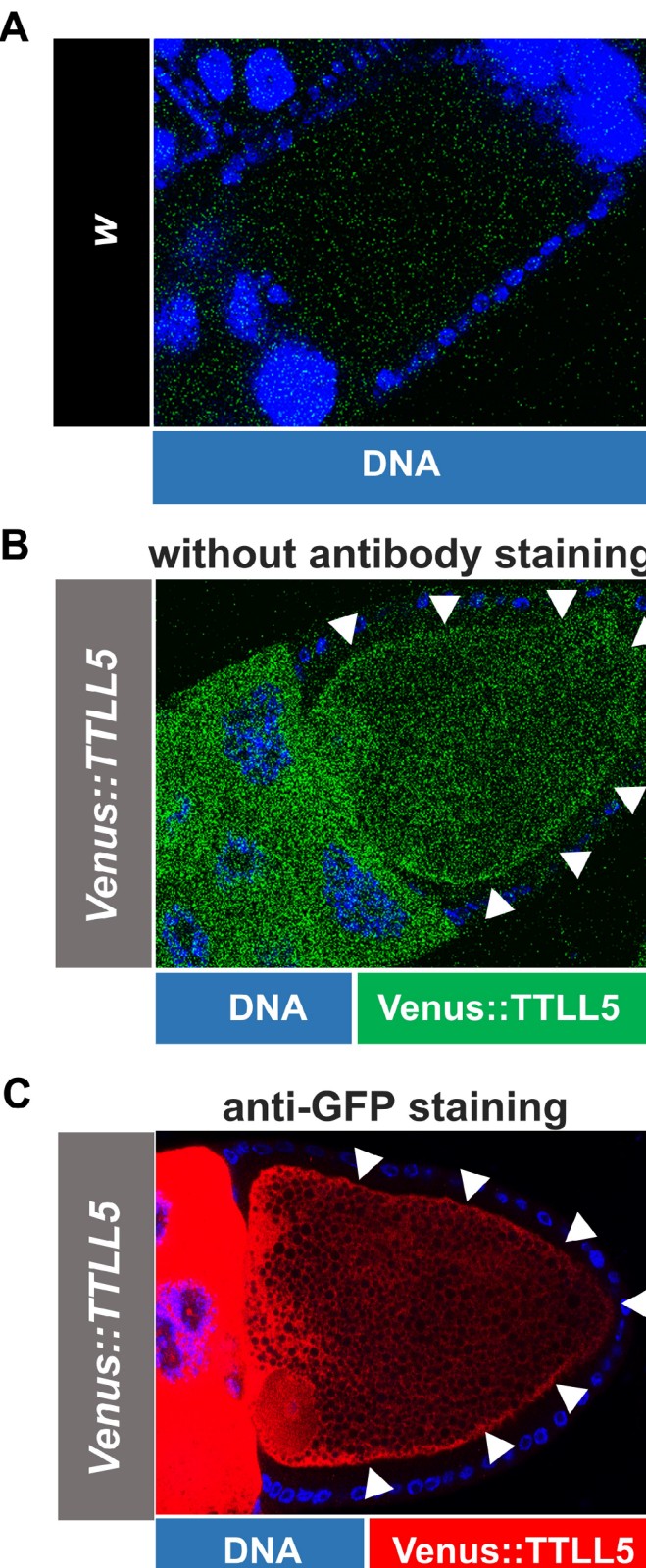

**Appendix 1—figure 3.** *Venus::TTLL5* expression in S10 oocytes. Confocal micrographs showing the S10 oocyte with or without anti-GFP staining. (**A**) *w* oocyte shows the autofluorescence background of the oocyte under the microscope. (**B**) The oocyte overexpressing *Venus:TTLL5* shows the live fluorescence of Venus (green). (**C**) The oocyte overexpressing *Venus:TTLL5* shows the Venus::TTLL5 protein after staining with an anti-GFP antibody (red). *Appendix 1—figure 3 continued on next page*

*Appendix 1—figure 3 continued*

Hoechst was in blue. The cortical signal of Venus::TTLL5 in the oocytes is pointed out with white arrowheads. The genotype for *Venus:TTLL5* overexpression was *MattubGal4/UAS-Venus::TTLL5*.

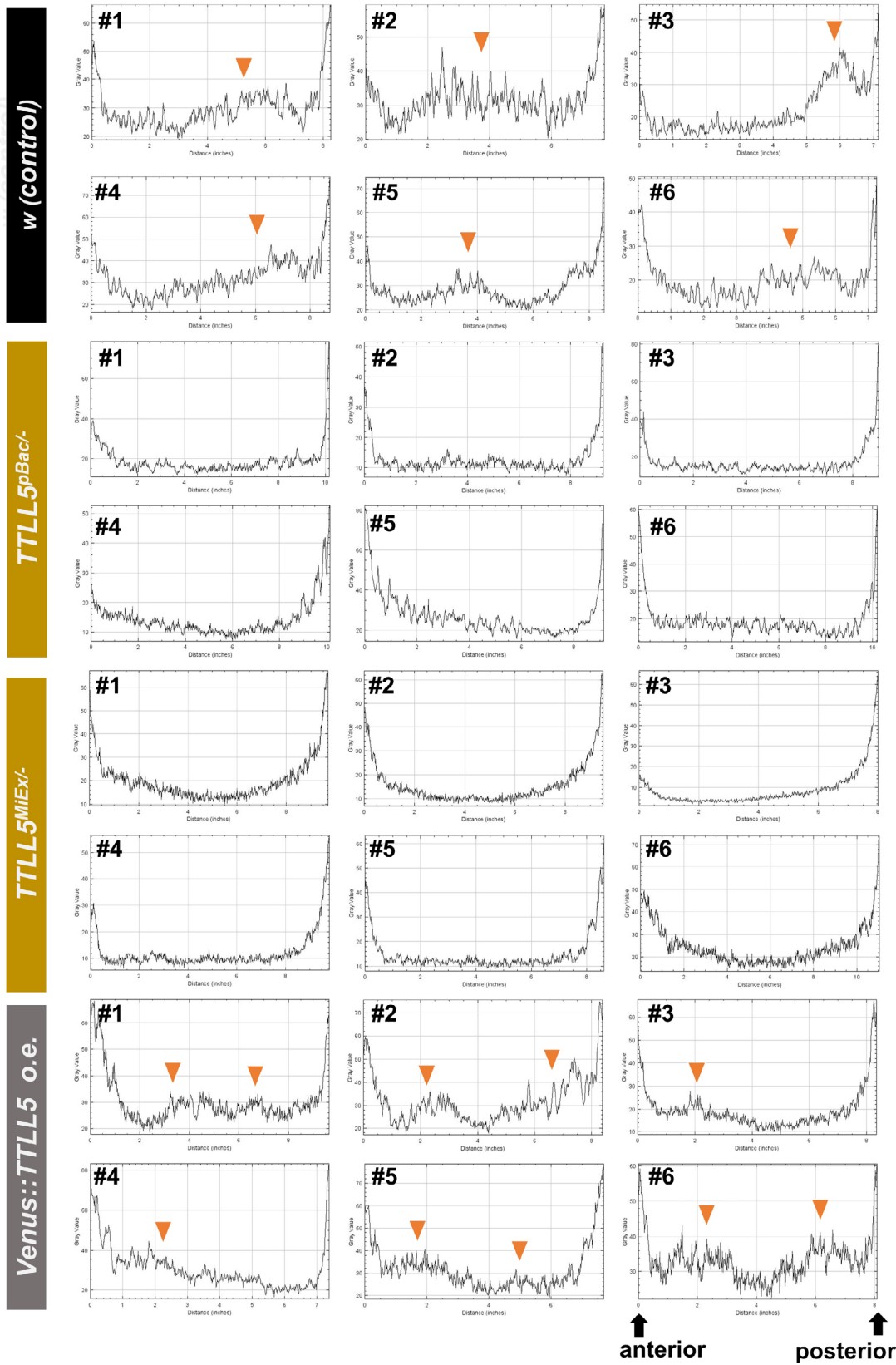

**Appendix 1—figure 4.** Khc distribution in S10B oocytes. Intensity charts from six samples of each genotype were plotted based on the line drawn along the AP axis of S10B oocytes (see *Figure 7E*). The orange triangles in the control and *o.e. Venus::TTLL5* point out the inner regions accumulating higher levels of Khc. The control was *w*

*Appendix 1—figure 4 continued on next page*

or *+/Df(3R)BSC679*. Genotypes for *TTLL5* mutants: *MattubGal4/+;TTLL5^pBac^/Df(3R)BSC679* and *TTLL5^MiEx^/Df(3R) BSC679*, respectively. Genotype for *TTLL5* overexpression: *MattubGal4/UAS-Venus::TTLL5*.

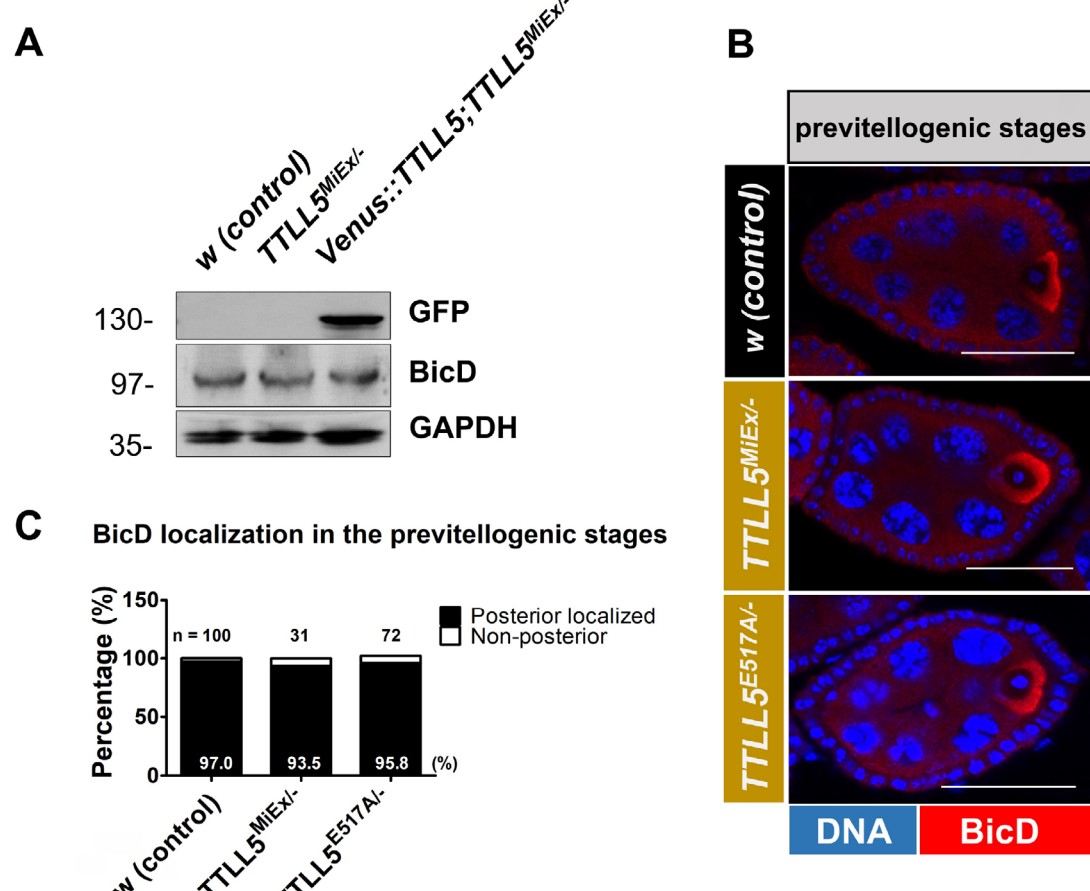

**Appendix 1—figure 5.** *TTLL5* has little or no effects on BicD expression and distribution in ovaries. (**A**) BicD levels remained unchanged between the *w* control, a *TTLL5* deficiency mutant, and the *TTLL5* rescue strain. GAPDH served as the loading control. (**B**) Confocal micrographs show BicD localizing preferentially at the posterior of the previtellogenic oocytes of *w* controls and *TTLL5* mutants. Scale bar: 25 µm. BicD protein in red, and Hoechst in blue. (**C**) 97%, 93.5%, and 95.8% of the oocytes showed posteriorly localized BicD in the *w* control, *TTLL5^MiEx^* and *TTLL5^E517A^* mutants, respectively. The total numbers of early oocytes are indicated as n = in the chart. Genotype for control: *w* or *+/Df(3R)BSC679*. Genotypes for *TTLL5* mutants: *TTLL5^MiEx^/Df(3R)BSC679* and *TTLL5^E517A^/Df(3R) BSC679*, respectively.

The online version of this article includes the following source data for appendix 1—figure 5:

**Appendix 1—figure 5—source data 1.** Source data for *Appendix 1—figure 5*.

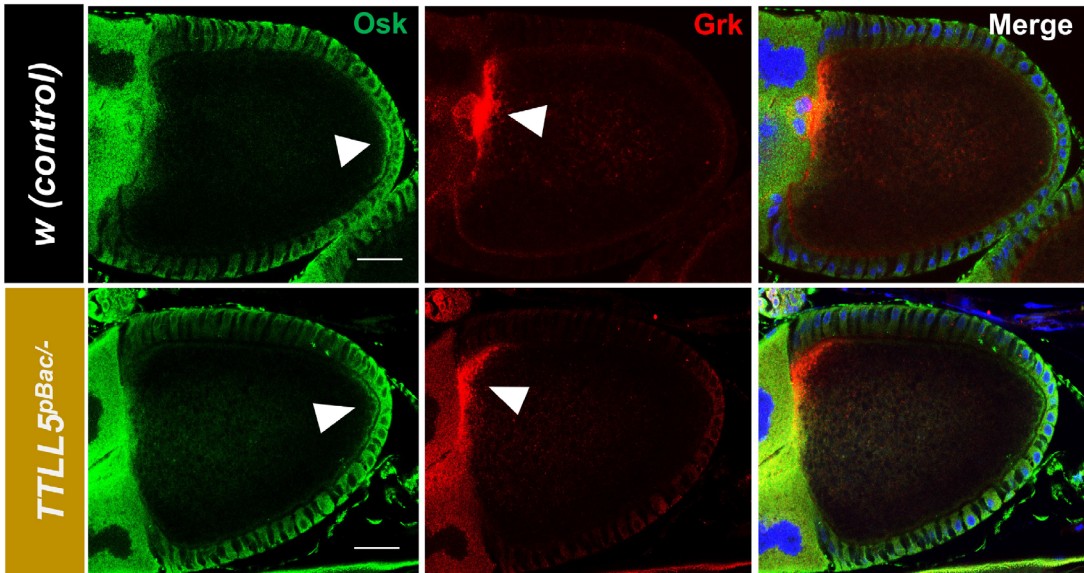

**Appendix 1—figure 6.** The polarity of S10B oocytes is normal in *TTLL5* mutant. In *TTLL5^pBac^* oocytes, Gurken (red) localizes properly to the dorsoanterior part of the oocyte cortex and Oskar (green) to the posterior cortex of 10B stage oocytes, indicating the polarity of MTs at this stage was normal. Blue channel: DNA. Genotypes for *TTLL5^pBac^*: *TTLL5^pBac^ /Df(3R)BSC679*.

