## [Editor Report]

Chemical modifications of the microtubule cytoskeleton regulate microtubule function in cells, but many open questions remain, such as the physiological relevance of these modifications and their impact on proteins that bind microtubules, such as molecular motors. In their useful study, the authors combine biochemistry, gene editing, and imaging to analyze one type of tubulin modification, tubulin glutamylation, which is enriched in developing ovaries and neurons. Their solid work implicates this modification in cytoplasmic streaming in oocytes and mitochondrial movement in neurons, which suggests that this modification may affect the activity of the molecular motor, kinesin-1. This work would be of interest to cellular and developmental biologists.

---

## [Decision Letter]

**Decision letter after peer review:**

[Editors’ note: the authors submitted for reconsideration following the decision after peer review. What follows is the decision letter after the first round of review.]

Thank you for submitting the paper "Differential modifications of the C-terminal tails of α-tubulin isoforms and their importance for kinesin-based microtubule transport in vivo" for consideration by *eLife*. Your article has been reviewed by 3 peer reviewers, and the evaluation has been overseen by a Reviewing Editor and a Senior Editor. The following individuals involved in review of your submission have agreed to reveal their identity: Francois Juge (Reviewer #3).

Comments to the Authors:

We are sorry to say that, after consultation with the reviewers, we have decided that this work will not be considered further for publication by *eLife*.

After a thorough discussion between all three reviewers, there was a consensus that this study clearly shows that TTLL5 acts as the primary writer for α-tubulin glutamylation, but the rest of the study falls short of providing a mechanistic effect of this post-translation modification. Although the authors correlate glutamylation with kinesin-1 transport, all three reviewers found that many further experiments would need to be performed to support this conclusion, especially since a prior paper, Devambez et al., 2017, did not report disruptions in vesicular transport upon TTLL5 loss-of-function.

*Reviewer #1 (Recommendations for the authors):*

In this manuscript, Bao et al. combine mutagenesis with mass spectrometry and western blotting to show that tubulin-tyrosine-ligase-like 5 (TTLL5) drives mono- and poly-glutamylation of the C-terminal ends of α-tubulin isoforms TBA1 and TBA3 in *Drosophila* ovaries. These findings advance our understanding of TTLL5 specificity in *Drosophila*. They then examine whether loss of glutamylation impairs kinesin-1-driven processes in the fly. They report altered Staufen localization in the oocyte and loss of directionality of ooplasmic streaming, as well as strongly diminished anterograde axonal trafficking of mitochondria in the wing nerves of mutant flies. These observations potentially represent an important contribution to our understanding of how tubulin glutamylation regulates motor-based transport in vivo.

Elements of the work are very well done, including comprehensive mutant and rescue data supporting the role of TTLL5 in glutamylation of TBA1/3 in the female germline. However, there are a number of substantive issues (particularly with data presentation and sample sizes) that need to be addressed before I could recommend that this data be published. Furthermore, the authors do not report the organismal effects of TTLL5 disruption, which is a further limitation of the study.

1. We have substantial concerns over the sample sizes for the wing nerve data presented in Figure 6 and Table 4. The authors observe a very strong reduction in kinesin-1-based transport in TTLL5 mutants. This result is very striking but needs to be confirmed with a sample size greater than three wings (there may be substantial wing-to-wing variability in transport parameters). Other groups typically analyze 6-10 wings for each condition. Similarly, the sample sizes of moving mitochondria used to quantify transport parameters are too low to draw strong conclusions, even if some of the statistical tests suggest significance (Table 4). Increasing sample sizes in the wing nerve assay may also allow the authors to determine if there is an effect on dynein-based motility, which would be a good addition to the paper. The authors should also examine mitochondrial transport in the TTLL5 mutants at a later time point (e.g. two days) to check if the mutants have a persistent problem in transport (rather than the timing of activation of the transport process).

2. The study would more impactful if the authors presented evidence that the cellular defects observed are accompanied by phenotypes at the organism level. For example, is there a neurodegenerative or oxidative stress phenotype in the mutant wing nerves, a reduction in female fertility, or a change in locomotion or lifespan of adult flies? Whilst some of these assays take a long time to complete, they have the potential to strengthen the manuscript substantially.

3. The authors should include a supplementary table clearly outlining the number of modified and unmodified peptides detected for both the wild-type and mutant mass spectrometry data to support their conclusions on the sites of glutamylation in TBA1/3, as well as their conclusion that TBA4 is not glutamylated in ovaries. The support for this latter conclusion could be weakened if there were far fewer peptides detected in total for TBA4 than 1/3.

4. The authors present fluorescent in situ hybridization (FISH) data that oskar mRNA is not mislocalized in TTLL5 mutant egg chambers during previtellogenic stages. It is essential to show FISH data for oskar mRNA localization in the later stage oocytes alongside the images of Staufen crescents (Figure 4A-B) to confirm that both are mislocalizing in the absence of glutamylation. Whilst Staufen is often used as a proxy for oskar mRNA, it is best to look at the mRNA directly -the authors presumably have the samples as they show oskar mRNA distribution in younger oocytes. Moreover, the localization of oskar and BicD mRNA in early oogenesis should be quantified to strengthen the conclusion that there is not a deficit in dynein-based transport.

5. The authors show evidence of impaired streaming in the mutant (Figure 5D). However, they also present data on streaming velocity in the mutants, which is unchanged compared to controls (Figure 5C). It is not clear how these values were determined, as there appear to be no discernable directional movements in the mutant examples in Figure 5D. The authors should clarify how the velocity values were obtained, and, if necessary, provide supplementary kymographs showing examples of movements in the mutant condition that were analyzed.

6. Based on the current data, the authors cannot rule out the effect on mitochondrial transport in the wing nerve of the TTLL5E517A mutant being due to the loss of protein function in other cells. Whilst this seems unlikely, the authors should acknowledge the possibility of an indirect effect when discussing their conclusions or testing this idea through a targeted RNAi approach.

7. The authors should take care to use appropriate language when interpreting their data; many of the stated conclusions are stronger than the evidence that supports them. For example, based on the effect of TTLL5E517A on kinesin-based mitochondrial transport in the wing, the authors conclude that "glutamylation of MTs is an essential regulatory mechanism" despite the fact that their data indicate that some transport can still be observed in regions of the wing. Similarly, the authors state in the discussion that "Our data show that TTLL5 is needed for … the posterior localization of Staufen/osk mRNA". However, Staufen does localize posteriorly in the mutant, and oskar is not assessed directly (see above). There are other instances of conclusions being worded too strongly so the authors need to go through the manuscript carefully with this in mind.

8. Lastly, we found the discussion to be largely functional and would be improved with a more in-depth consideration of the literature throughout. In particular, greater emphasis and discussion are needed on their transport data in the context of the contrasting findings of Devambez et al., 2017 and Bodakuntla et al., 2021. We also feel the discussion would benefit from a concluding section outlining the broader implications of these results for the field and future perspectives for the work.

*Reviewer #2 (Recommendations for the authors):*

The authors' manuscript very nicely describes the glutamylation of α-tubulin in *Drosophila* in ovaries and the effects of eliminating the glutamylase TTLL5 on tubulin glutamylation. Overall, the data support the conclusion that TTLL5 is responsible for α-tubulin glutamylation, although some of the western blots have multiple bands that make the data a bit difficult to interpret. The use of specific point mutations in TTLL5 is a strength. The authors' data indicating that loss of TTLL5 disrupts both cytoplasmic streaming in oocytes and mitochondrial movement in neuronal axons are also generally convincing, albeit with some comments and concerns. Since both processes involve kinesin-1, the authors go on to propose that TTLL5-mediated glutamylation of α-tubulin regulates kinesin-1. This model is logically based on the authors' data and published work, but there is no direct support for this conclusion, and the mechanism by which glutamylation would affect kinesin-1 (transport, microtubule sliding) is fuzzy and speculative. Thus, this study is essentially a descriptive report of TTLL5.

1. The model that TTLL5-mediated tubulin glutamylation regulates kinesin-1 is not supported. Have the authors tested for a genetic interaction between TTLL5 and kinesin-1 in oocytes and neurons? The kinesin-1 mutation that disrupts microtubule sliding and cytoplasmic streaming in oocytes (mutA mutation) would likely be quite helpful, too. Regarding oocytes: Although the levels of kinesin-1 are not affected, is its distribution affected? Does TTLL5 affect the localization of kinesin-1 or just its activity? What might this mean regarding a model of how TTLL5 (and tubulin glutamylation) affect cytoplasmic streaming?

2. Kinesin-1's microtubule sliding activity is thought to be separable from its cargo transport activity; it is not clear how microtubule glutamylation would affect both these processes. The authors should take care to discriminate between kinesin-1-mediated microtubule sliding activities, such as cytoplasmic streaming, versus its transport activities, such as mitochondrial movement.

3. What is the distribution of glutamylated microtubules and TTLL5 in oocytes? Where would TTLL5 be acting? If TTLL5 has a localized distribution and/or activity, what would this mean with regards to how TTLL5 affects cytoplasmic streaming?

4. Are oocyte microtubules affected by the loss of TTLL5? By the over-expression of TTLL5?

5. A previous publication (Devambez et al. 2017, referenced by the authors) has reported that loss of TTLL5 does not affect vesicle transport. Is the effect of glutamylation specific to mitochondria movement? The authors should determine whether vesicle transport is affected or not in their system, and determine whether glutamylation may have cargo-specific effects (e.g. see Zheng et al. 2022, PMID: 34912111).

6. The authors describe reports from mammalian models that increased glutamylation, rather than loss of glutamylation, may be disruptive. What are the effects of increasing TTLL5 in oocytes and/or neurons in flies?

*Reviewer #3 (Recommendations for the authors):*

The authors explore the potential role of α-tubulin polyglutamylation during oogenesis and in mitochondria transport into neurons. In ovaries, they identify polyglutamylation sites within the tail of TBA1/3 isotypes but surprisingly not within the tail of the female germline-specific TBA4 isotype. They identify TTLL5 as the enzyme responsible for polyglutamylation of TBA1/3 in ovaries and thoracic neurons using a series of TTLL5 mutants, including loss-of-function and enzymatically dead versions engineered by CRISPR. They show that TTLL5 mutations affect cytoplasmic streaming in oocytes and the narrowed localization of Staufen to the posterior pole of the oocyte, two phenotypes that are rescued by expression of TTLL5 in the germline. Finally, the authors report that enzymatically dead TTLL5 mutants affect the transport of mitochondria in wing nerves.

The data showing that TTLL5 is involved in oocyte cytoplasmic streaming and Staufen localization are convincing. Nonetheless, the authors do not address if tubulin polyglutamylation occurs in the germline, in the somatic tissues of the ovaries, or both. The fact that the female germline-specific TBA4 isotype is not glutamylated could argue in favor of glutamylation occurring mainly in the somatic cells. The lower amount of TBA4 compared to TBA1/3 detected by MS also reflects the high quantity of tubulin coming from follicle cells, which may mask glutamylation of TBA4.

Since the contribution of kinesin 1 to the TTLL5 mutant phenotypes in oocytes or neurons has not been addressed, it is premature to conclude that these phenotypes are mediated by an effect of microtubules polyglutamylation on kinesin-based transport.

The authors do not comment on the fact that a previously published study (Devambez et al. 2017) did not detect any effect of TTLL5 loss-of-function on vesicular transport. The discrepancies could be due to the experimental system or more likely to the nature of the TTLL5 alleles used in the present study which are catalytically-dead enzymes that may act as dominant negatives. It would be interesting to compare the different mutants in the same system and to rescue the transport phenotype with wild-type TTLL5, to firmly establish the role of TTLL5 in mitochondrial transport.

The authors also report the presence of polyglycylated α-tubulin in ovaries, detected by mass spectrometry. This observation, which is unexpected given that the known *Drosophila* polyglycylases (TTLL3A/B) are not expressed in ovaries, is potentially interesting but should be confirmed by western blotting.

It would be meaningful to analyze by immunostaining the distribution of polyglutamylated microtubules (MTs) in ovaries to determine if this modification occurs in follicle cells, oocytes, or both. One easy way to demonstrate that polyglutamylation occurs in the germline could be to express a GFP-tagged α-tubulin using the nos-gal4 driver and analyze by western blotting if this protein is polyglutamylated. This approach could also allow determining if the TUBA4 isotype can be polyglutamylated.

Another approach could be to make TTLL5 mutant clones in the germline by FRT-mediated recombination and analyze their phenotype (cytoplasmic streaming and localization of Staufen). This would demonstrate that the phenotypes are not due to defects in follicle cells and would allow the analysis of the MT network in TTLL- oocyte. Does Staufen colocalize with glutamylated MTs in wild-type oocytes?

The existence of polyglycylated tubulin in the female germline should be confirmed by western blotting using specific antibodies that are available.

It is important to address the differences between the present study and the results from Devambez et al. Do the catalytically dead enzymes behave as dominant negative? Are the transport phenotypes observed in TTLL5E517A rescued by expression of Venus-TTLL5? Are they observed in TTLL5E517A/+ flies?

I have concerns regarding the signal detected by β-monoE antibody in ovaries. According to the initial publication (Bodakuntla et al. 2021), the β-monoE antibody recognizes glutamylated GEF motif only, but this sequence is found only in the tail of βTub85D isotype which is testis specific. Did the authors address the specificity of this new antibody in *Drosophila*? Did the author analyze glutamylation sites of βTubulins by mass spectrometry?

In the text, "tubulin isoforms" should be changed to "tubulin isotypes" since they are produced from different genes.

[Editors’ note: further revisions were suggested prior to acceptance, as described below.]

Thank you for resubmitting your work entitled "Differential modification of the C-terminal tails of different α-tubulins and their importance for microtubule function in vivo" for further consideration by *eLife*. Your revised article has been evaluated by Anna Akhmanova (Senior Editor) and a Reviewing Editor.

The manuscript has been improved but there are some remaining issues that need to be addressed, as outlined below:

The reviewers were split upon re-review of this work. There were concerns that the TTLL4 data were too preliminary and do not strengthen the paper and should be removed. In addition, all reviewers agreed in the discussion that the speculation about microtubule severing should be removed entirely. One reviewer pointed out that the authors use the example of spastin, but spastin-mediated severing is mainly affected by polyglutamylation on β-tubulin, which is not the case in this paper. The general consensus in the review discussion is that there is still a lack of experimental support for a direct effect of microtubule glutamylation on kinesin-1 activity. Therefore, the suggestion is for the authors to soften the claims and provide a sentence or two about how future in vitro studies will be necessary to determine if there is a direct effect on the motor itself or if there are effects on other MAPs, enzymes or processes that could produce the in vivo results.

*Reviewer #1 (Recommendations for the authors):*

The authors set out the examine the in vivo function of glutamylation of the C-terminal tails of microtubules. Tubulin glutamylation is widely observed but the precise tubulin isotypes that undergo this modification and the in vivo consequences of these changes are not well understood. The authors provide evidence that specific tubulin isotypes undergo this modification and that these events influence kinesin-1 function using well established model systems in oogenesis and in wing nerve axons. This work complements and extends previous studies of the function of glutamylation of microtubules in vivo. Strengths of the study include painstaking generation and characterization of several new alleles in the TTLL5 glutamylase. The use of multiple alleles strengthen the conclusions drawn. These reagents will also be valuable for other researchers. The conclusions are supported by quantitative analysis, including of several readouts of kinesin-1 activity. Whilst the authors do not address how polyglutamylation affects kinesin-1 function, the work will stimulate other efforts to address this issue.

The authors have satisfactorily addressed most of my comments. In particular, the wing nerve analysis is much improved and has led to interesting conclusions. The revisions in response to the other reviewers have also led to significant improvements. Whilst the function of microtubule glutamylation has been assessed in other systems, the work represents a valuable addition and also generates and characterizes useful reagents for the community.

However, I am still confused about some of the mass spectromery data. Whilst it seems clear from the rebuttal and revision that atub84B/D has a much higher frequency of glutamylation than atub67C, I do not understand which data support the conclusions in Figure 1. Is this a separate experiment to the one documented in Table 2 and S2? If so, the data should also be presented or the experiment leading to Figure 1 not mentioned (if the conclusions are fully supported by the later experiment). If all these data come from the same experiment, this issue should be clarified and the tables called out before Figure 1.

The evidence supporting the conclusion that tub84B/D is differentially glutamylated vs tub67C would be easier to extract if the percentage modification and total number of peptides was additionally stated for both categories in the text.

*Reviewer #2 (Recommendations for the authors):*

In cells, microtubules acquire chemical modifications that are thought to affect the interactions between microtubules and microtubule-binding proteins. The authors address the question of what role microtubule glutamylation may have on microtubule function in cells by manipulating the enzyme that glutamylates microtubules and using *Drosophila* as a model.

A strength of the manuscript is the authors' in vivo analysis of the TTLL5 enzyme that glutamylates microtubules. The mutations that were introduced into endogenous TTLL5 create a useful toolbox for the in vivo analysis of TTLL5 and microtubule glutamylation. Also, in general, the data are clearly presented and are consistent with the authors conclusions. Major weaknesses of the manuscript, however, are that (1) the authors' central model that microtubule glutamylation affects kinesin-1-dependent processes is supported by correlative data but not any functional tests, and (2) the phenotypes resulting from the loss of microtubule glutamylation are quite modest.

This study is a nicely done characterization of a limited suite of TTLL5 phenotypes in an in vivo model (fly oocyte), and the authors' description of the effects of the loss of microtubule glutamylation will be of interest to researchers working in the field of microtubule post-translational modifications.

A central challenge in the field of microtubule post-translational modifications (PTMs) is to determine the physiological significance of these PTMs and the mechanism by which they may act. The authors generate useful TTLL5 reagents and characterize the effects of disrupting microtubule glutamylation in oocytes (and neurons) in vivo, but these observations by themselves do not provide sufficient support to the authors' central model that microtubule glutamylation affects kinesin-1-mediated processes. The authors have generated new tools and information that will be useful to the field, but the modest phenotypes and lack of functional tests between TTLL5 and kinesin-1 limit the conceptual and scientific advance.

Overall, this is a solid but limited characterization of TTLL5 mutants. My major concerns remain about the model connecting microtubule glutamylation and kinesin-1: this model is based on correlation and somewhat modest phenotypes. There is also quite a bit of speculation (the introduction of microtubule severing into a potential mechanism seems very speculative). Some of my original major concerns were not addressed in the revision. I do not support publication.

*Reviewer #3 (Recommendations for the authors):*

Bao et al. investigate the function of Tubulin glutamylation in *Drosophila* oocyte and in mitochondria transport in adult nerve. Using Mass Spectrometry (MS), they detect glutamylation only on αTub84B/D C-terminal tails but not on the maternal αTub67C isotype. αTub84B/D are mainly monoglutamylated and polyglutamylation does not extend more than 3 glutamates. They identify TTLL5 as a major glutamylase during oogenesis, using previously described mutants as well as new point mutation alleles produced by CRISPR. In TTLL5 mutant backgrounds, cortical localization of Staufen at the posterior pole of S10 oocytes is more diffuse, and cytoplasmic streaming is abnormal. Significantly, these phenotypes are partially rescued by expression of a Venus-TTLL5 construct. The authors show that localization of Khc in S10 oocytes is also affected by loss-of-function as well as upregulation of TTLL5. Finally, they show that mitochondria transport in the wing nerve is affected in TTLL5 mutants background in which pausing time is decreased in anterograde but not in retrograde transport. Altogether these data suggest a link between microtubules glutamylation mediated by TTLL5 and kinesin1-dependent processes.

In this revised version of the manuscript the authors took into account most criticisms of the reviewers and overall, the manuscript has been improved. Some data and conclusions that looked preliminary in the first version have been completely deleted. On the other hand, new data and quantifications, especially in Figures 6, 7 and 8, now strengthen the results and conclusions of the authors. Presentation of the results has also been improved. However, new data concerning the potential role of TTLL4A as a glutamylase in the ovary are preliminary and miss important controls. The relative contribution of TTLL4A and TTLL5 to αTub84B/D glutamylation is not explored.

I have concerns about the data on TTLL4A. Indeed, the authors now explored the function TTLL4A (Figure S2). It is not clear why they initially supposed that TTLL4A could be involved in glycylation of αTubulin. The presence of glycylation in ovaries is not supported by strong data and the authors recognize that it could be due to an artifact of the MS procedure itself. Moreover, one cannot rule out that detection of low glycylation levels on αTubulin by MS is due to a contamination of the ovaries sample by sperm present in spermateca. Indeed, the polyglycylation signal is extremely strong on sperm axoneme. Analyzing the ovaries of virgin females would definitely prove that glycylation is present or not during oogenesis.

More importantly, there is not sufficient data to assure that TTLL4A is also involved in glutamylation of αTubulin. First, the mutants used are not characterized. At least RT-PCR is necessary to show that the expression level of TTLL4A is decreased in the mutant contexts used. Moreover, no rescue of the phenotype is presented. It seems important to note that the genomic deficiency used, which includes TTLL4, also deletes piwi and aubergine, two genes which have important roles during oogenesis. It is therefore necessary to rule out that the transposons insertions in the TTLL4 gene do not impact the expression of a nearby gene. In the western blots all genotypes should be probed with both GT335 and 1D5 as well as with a Tubulin antibody as loading control.

Finally, the impact of TTLL4 depletion on tubulin glutamylation, as presented, is as strong as the impact of TTLL5 (Figure S2). How do the authors reconcile this result with their observation that no glutamylation is detected in TTLL5 mutant by MS? What is the relative contribution of TTLL4A and TTLL5 to αTub84B/D glutamylation? Using RNAi-mediated depletion of TTLL4A and TTLL5 would be a simple strategy to address these questions.

Overall, the data with TTLL4 are too preliminary and do not strengthen the conclusions of the paper.

In most figures, the annotation of the genotypes is misleading since TTLL5 mutants are used as heterozygous over a deficiency. This is also not mentioned in the main text but only in the figure legends. The genotypes are presented as TTLL5/- in figure 2 but not in other figures. The presentation should be consistent throughout the figures and clearly mentioned in the main text. Is there a stronger phenotype in TTLL5-/Def compared to TTLL5-/- homozygous? Why only hemizygous contexts were used?

Figure 6 The number of oocytes (7) for the rescue experiment is too low to draw the conclusion that streaming is restored in the majority of oocytes.

---

## [Author Response]

[Editors’ note: the authors resubmitted a revised version of the paper for consideration. What follows is the authors’ response to the first round of review.]

Comments to the Authors:We are sorry to say that, after consultation with the reviewers, we have decided that this work will not be considered further for publication by eLife.After a thorough discussion between all three reviewers, there was a consensus that this study clearly shows that TTLL5 acts as the primary writer for α-tubulin glutamylation, but the rest of the study falls short of providing a mechanistic effect of this post-translation modification. Although the authors correlate glutamylation with kinesin-1 transport, all three reviewers found that many further experiments would need to be performed to support this conclusion, especially since a prior paper, Devambez et al., 2017, did not report disruptions in vesicular transport upon TTLL5 loss-of-function.Reviewer #1 (Recommendations for the authors):In this manuscript, Bao et al. combine mutagenesis with mass spectrometry and western blotting to show that tubulin-tyrosine-ligase-like 5 (TTLL5) drives mono- and poly-glutamylation of the C-terminal ends of α-tubulin isoforms TBA1 and TBA3 in *Drosophila* ovaries. These findings advance our understanding of TTLL5 specificity in *Drosophila*. They then examine whether loss of glutamylation impairs kinesin-1-driven processes in the fly. They report altered Staufen localization in the oocyte and loss of directionality of ooplasmic streaming, as well as strongly diminished anterograde axonal trafficking of mitochondria in the wing nerves of mutant flies. These observations potentially represent an important contribution to our understanding of how tubulin glutamylation regulates motor-based transport in vivo.Elements of the work are very well done, including comprehensive mutant and rescue data supporting the role of TTLL5 in glutamylation of TBA1/3 in the female germline. However, there are a number of substantive issues (particularly with data presentation and sample sizes) that need to be addressed before I could recommend that this data be published. Furthermore, the authors do not report the organismal effects of TTLL5 disruption, which is a further limitation of the study.1. We have substantial concerns over the sample sizes for the wing nerve data presented in Figure 6 and Table 4. The authors observe a very strong reduction in kinesin-1-based transport in TTLL5 mutants. This result is very striking but needs to be confirmed with a sample size greater than three wings (there may be substantial wing-to-wing variability in transport parameters). Other groups typically analyze 6-10 wings for each condition. Similarly, the sample sizes of moving mitochondria used to quantify transport parameters are too low to draw strong conclusions, even if some of the statistical tests suggest significance (Table 4). Increasing sample sizes in the wing nerve assay may also allow the authors to determine if there is an effect on dynein-based motility, which would be a good addition to the paper. The authors should also examine mitochondrial transport in the TTLL5 mutants at a later time point (e.g. two days) to check if the mutants have a persistent problem in transport (rather than the timing of activation of the transport process).

We are grateful for the suggestion to analyze the mitochondrial transport in the wing nerves at a slightly later time in development (24-48 hours after eclosion) because the previously observed effects might be caused by a slight developmental delay. We also increased the sample size to 7-8 flies/wings for each genotype as suggested. With this setup, we obtained consistent results. Antero- and retrograde transport of mitochondria took place, but in the absence of TTLL5 enzymatic activity, the kinesin-dependent anterograde mitochondrial transport paused less frequently, while the dynein-based motility seemed not to be affected.

2. The study would more impactful if the authors presented evidence that the cellular defects observed are accompanied by phenotypes at the organism level. For example, is there a neurodegenerative or oxidative stress phenotype in the mutant wing nerves, a reduction in female fertility, or a change in locomotion or lifespan of adult flies? Whilst some of these assays take a long time to complete, they have the potential to strengthen the manuscript substantially.

Regarding the impact of lack of glutamylation at the animal level, Devambez et al. (2017) already reported their analyses of several assays at this level. This includes larval neuromuscular junction morphology, vesicular axonal transport in larval nerves, larval locomotion behavior, and adult life span. In many cases, they observed differences between the mutant and the control that showed a very mild expected effect, but they were not significant. In our assays to test female fertility, the lack of *TTLL5* neither caused a significant decrease in hatching rates, nor distinguishable morphological changes in ovaries. The lack of glutamylated tubulin generally seems to stress the affected cells, but many cells can compensate for it. Such effects can be proven to be significant only with very laborious work (e.g.: by studying the behavior of aged flies using genetic backgrounds that were matched with the controls through repeated backcrossing). Instead of going this route, we have focused on those cellular events where we can directly visualize the role of glutamylation (even if this role is not essential for viability or fertility). Additionally, in the new version, we now elaborate a bit more on the results of Devambez et al. (2017).

3. The authors should include a supplementary table clearly outlining the number of modified and unmodified peptides detected for both the wild-type and mutant mass spectrometry data to support their conclusions on the sites of glutamylation in TBA1/3, as well as their conclusion that TBA4 is not glutamylated in ovaries. The support for this latter conclusion could be weakened if there were far fewer peptides detected in total for TBA4 than 1/3.

Most of this information had already been listed in Tables 2 and S2. Table 2 shows that 62% (35 peptides of a total of 56) of the αTub84B/D (=TBA1/3) peptides detected contained Glu modifications in the wild type. That 0% αTub84B/D peptides were modified in the *TTLL5* mutants is also indicated in Table 2 and the text. The total number of C-terminal peptides is seen in the footnote of Table 2. Because αTub67C (TBA4) modification was neither observed in the wild type nor in the mutants, we only mention this in the text. We, therefore, do not see the need for an additional Table and simply modified the existing Table 2 by more clearly indicating the “(%E mod)” in Table 2 and by revising the footnote. There are fewer peptides for the mutants, but because 62% are modified in the wild type (n=56), the 0% of 36 mutant peptides should be sufficient to claim that the TTLL5 is required for this glutamylation. Regarding the lack of modification aTub67C, we have analyzed 12 C-terminal peptides from the wild type and 16 from the *TTLL5* mutants and have not observed any glutamylation. In this case, there is more of a chance that we missed a rare modification.

We, therefore, point this out in the manuscript as follows:

“Because of the lower abundance of Tub67C C-terminal peptides, we cannot rule out that rare glutamylation on the maternal αTub67C exists. Altogether, we conclude that

glutamylation of α-tubulin had a strong preference for the C-terminal domain of αTub84B/D.”

4. The authors present fluorescent in situ hybridization (FISH) data that oskar mRNA is not mislocalized in TTLL5 mutant egg chambers during previtellogenic stages. It is essential to show FISH data for oskar mRNA localization in the later stage oocytes alongside the images of Staufen crescents (Figure 4A-B) to confirm that both are mislocalizing in the absence of glutamylation. Whilst Staufen is often used as a proxy for oskar mRNA, it is best to look at the mRNA directly -the authors presumably have the samples as they show oskar mRNA distribution in younger oocytes. Moreover, the localization of oskar and BicD mRNA in early oogenesis should be quantified to strengthen the conclusion that there is not a deficit in dynein-based transport.

We now removed the *osk* and *BicD* mRNA data because we did not have enough data. Instead, we have included Staufen protein localization during the early and mid-stage oocytes (Figure 5F). It appears that the localization pattern of Staufen in these stage oocytes is not affected by *TTLL5* mutants. Being the adaptor for the Dynein-dependent transport machinery, BicD is even closer to assessing the dynein-based transport than *osk* mRNA. Therefore, we have now also included and quantified the BicD protein localization in earlystage oocytes to strengthen the conclusion that the dynein-dependent transport does not appear to be affected by TTLL5 (Figure S5) and that the orientation of the MTs is normal.

5. The authors show evidence of impaired streaming in the mutant (Figure 5D). However, they also present data on streaming velocity in the mutants, which is unchanged compared to controls (Figure 5C). It is not clear how these values were determined, as there appear to be no discernable directional movements in the mutant examples in Figure 5D. The authors should clarify how the velocity values were obtained, and, if necessary, provide supplementary kymographs showing examples of movements in the mutant condition that were analyzed.

We have decided to remove the streaming velocity data because the differences in streaming patterns prevent a meaningful interpretation of such data. Instead, we now focus on the different streaming patterns seen in the different genotypes and illustrate them using cartoons. We also quantified the appearance of these patterns for each genotype (Figure 6C and 6D). As suggested, we now provide two kymographs for each genotype, and we additionally included the streaming patterns in oocytes where TTLL5 was overexpressed to provide a more comprehensive picture of the impact of tubulin-glutamylation on ooplasmic streaming.

6. Based on the current data, the authors cannot rule out the effect on mitochondrial transport in the wing nerve of the TTLL5E517A mutant being due to the loss of protein function in other cells. Whilst this seems unlikely, the authors should acknowledge the possibility of an indirect effect when discussing their conclusions or testing this idea through a targeted RNAi approach.

We agree that we did not show that this is a cell-autonomous effect and also agree that a cell-autonomous effect is very likely. We have now added this point at the end of the results on page 13.

7. The authors should take care to use appropriate language when interpreting their data; many of the stated conclusions are stronger than the evidence that supports them. For example, based on the effect of TTLL5E517A on kinesin-based mitochondrial transport in the wing, the authors conclude that "glutamylation of MTs is an essential regulatory mechanism" despite the fact that their data indicate that some transport can still be observed in regions of the wing. Similarly, the authors state in the discussion that "Our data show that TTLL5 is needed for … the posterior localization of Staufen/osk mRNA". However, Staufen does localize posteriorly in the mutant, and oskar is not assessed directly (see above). There are other instances of conclusions being worded too strongly so the authors need to go through the manuscript carefully with this in mind.

Thanks for the suggestions. We made an effort to be precise in our statements and not to over-state the effect of TTLL5.

8. Lastly, we found the discussion to be largely functional and would be improved with a more in-depth consideration of the literature throughout. In particular, greater emphasis and discussion are needed on their transport data in the context of the contrasting findings of Devambez et al., 2017 and Bodakuntla et al., 2021. We also feel the discussion would benefit from a concluding section outlining the broader implications of these results for the field and future perspectives for the work.

We largely refrained from discussing the function of the mammalian *TTLL5* (and also proteins whose homologs are not expressed in fly ovaries). In light of the new results and the encouragement of the reviewers, we now discuss the possible mechanism of the effect of glutamylated tubulins on kinesin transport. In this discussion, we now also cover the new papers that link glutamylation to MT severing and we suggest that glutamylation may interrupt the transport and/or MT sliding activity of kinesin-1 by severing MTs.

Additionally, in the new version, we have also discussed the point regarding the impact of lack of glutamylation on vesicle transport in Devambez et al., 2017. These authors actually observed differences between the mutant and the control that showed the expected effect, but the differences were not statistically significant*.* Also, our new results are more in line with the effects described by Bodakuntla et al., 2021.

Reviewer #2 (Recommendations for the authors):The authors' manuscript very nicely describes the glutamylation of α-tubulin in *Drosophila* in ovaries and the effects of eliminating the glutamylase TTLL5 on tubulin glutamylation. Overall, the data support the conclusion that TTLL5 is responsible for α-tubulin glutamylation, although some of the western blots have multiple bands that make the data a bit difficult to interpret. The use of specific point mutations in TTLL5 is a strength. The authors' data indicating that loss of TTLL5 disrupts both cytoplasmic streaming in oocytes and mitochondrial movement in neuronal axons are also generally convincing, albeit with some comments and concerns. Since both processes involve kinesin-1, the authors go on to propose that TTLL5-mediated glutamylation of α-tubulin regulates kinesin-1. This model is logically based on the authors' data and published work, but there is no direct support for this conclusion, and the mechanism by which glutamylation would affect kinesin-1 (transport, microtubule sliding) is fuzzy and speculative. Thus, this study is essentially a descriptive report of TTLL5.1. The model that TTLL5-mediated tubulin glutamylation regulates kinesin-1 is not supported. Have the authors tested for a genetic interaction between TTLL5 and kinesin-1 in oocytes and neurons? The kinesin-1 mutation that disrupts microtubule sliding and cytoplasmic streaming in oocytes (mutA mutation) would likely be quite helpful, too. Regarding oocytes: Although the levels of kinesin-1 are not affected, is its distribution affected? Does TTLL5 affect the localization of kinesin-1 or just its activity? What might this mean regarding a model of how TTLL5 (and tubulin glutamylation) affect cytoplasmic streaming?

As recommended, in the new version we strengthened the proposed pathway of action for αtubulin glutamylation by showing that it affects the distribution of Kinesin-1, particularly it’s accumulation at the cell cortex (Figure 7 and S4). In addition, we show that glutamylation, too, is normally seen preferentially at the cortex of the oocyte (Figure 4). Furthermore, both cortical localization patterns depend on *TTLL5* activity. We also discuss the interesting coincidence that the proper streaming pattern and the proper Kinesin-1 distribution both require moderate levels of active TTLL5. Together with the discussion of the new literature about the role of glutamyl side chains in recruiting MT severing proteins, our new results provide now interesting insights into the involvement of Kinesin-1 in the TTLL5-mediated cytoplasmic streaming.

2. Kinesin-1's microtubule sliding activity is thought to be separable from its cargo transport activity; it is not clear how microtubule glutamylation would affect both these processes. The authors should take care to discriminate between kinesin-1-mediated microtubule sliding activities, such as cytoplasmic streaming, versus its transport activities, such as mitochondrial movement.

Considering the new results from the literature (role of glutamylation in severing MTs) and our data described in the response to the previous point, the primary effect could also be similar. Severed MTs might prevent the movement of kinesin-1. Alternatively, or additionally, given the results from Lu et al., it seems also possible that the MT binding site in the coiled coil region of KHC (specifically required for the MT sliding) has a different affinity to glutamylated tubulin tails. We now discuss the possible mechanisms of the effect of glutamylated tubulins on the kinesin-1 activities in the discussion.

3. What is the distribution of glutamylated microtubules and TTLL5 in oocytes? Where would TTLL5 be acting? If TTLL5 has a localized distribution and/or activity, what would this mean with regards to how TTLL5 affects cytoplasmic streaming?

As we stated in the response to question 1, we observed glutamylation preferentially at the cortex of the oocyte (Figure 4). Coincidently, maternally (over-)expressed Venus::TTLL5 is also enriched at the cortex region (Figure S3), similar to Khc and glutamylated MTs, suggesting a potential correlation between glutamylated cortical MTs and ooplasmic streaming. In addition, by examing the distribution of kinesin-1, we have uncovered a small additional central peak of Khc in the oocyte. Whether this could contribute to the streaming or is “only” the consequence of the ooplasmic streaming still needs to be shown.

4. Are oocyte microtubules affected by the loss of TTLL5? By the over-expression of TTLL5?

We have not observed changes in microtubule organization by either loss of TTLL5 or overexpression of TTLL5 (but we observed loss of the polyglutamyl signal, suggesting that their modification is affected). We now show that the orientation of the oocyte MTs is normal upon loss of *TTLL5* (Figure S6).

5. A previous publication (Devambez et al. 2017, referenced by the authors) has reported that loss of TTLL5 does not affect vesicle transport. Is the effect of glutamylation specific to mitochondria movement? The authors should determine whether vesicle transport is affected or not in their system, and determine whether glutamylation may have cargo-specific effects (e.g. see Zheng et al. 2022, PMID: 34912111).

Devambez et al., 2017 actually observed expected differences regarding the pausing of the vesicle transport in larval neurons, but these differences between the mutant and the control were not statistically significant. We think that by investigating cargo transport in neurons using a different system, we obtained more consistent results with statistically significant differences.

6. The authors describe reports from mammalian models that increased glutamylation, rather than loss of glutamylation, may be disruptive. What are the effects of increasing TTLL5 in oocytes and/or neurons in flies?

To also provide the analysis of the TTLL5 overexpression phenotype, we now added the analyses of the cytoplasmic streaming in oocytes that overexpress TTLL5. Increasing TTLL5 does not appear to affect the microtubule organization in oocytes, female fertility, or wing morphology. However, over-expression of TTLL5 does affect (somewhat) ooplasmic streaming patterns and kinesin-1 distribution at the cellular level in oocytes (Figure 6 and 7).

Reviewer #3 (Recommendations for the authors):The authors explore the potential role of α-tubulin polyglutamylation during oogenesis and in mitochondria transport into neurons. In ovaries, they identify polyglutamylation sites within the tail of TBA1/3 isotypes but surprisingly not within the tail of the female germline-specific TBA4 isotype. They identify TTLL5 as the enzyme responsible for polyglutamylation of TBA1/3 in ovaries and thoracic neurons using a series of TTLL5 mutants, including loss-of-function and enzymatically dead versions engineered by CRISPR. They show that TTLL5 mutations affect cytoplasmic streaming in oocytes and the narrowed localization of Staufen to the posterior pole of the oocyte, two phenotypes that are rescued by expression of TTLL5 in the germline. Finally, the authors report that enzymatically dead TTLL5 mutants affect the transport of mitochondria in wing nerves.The data showing that TTLL5 is involved in oocyte cytoplasmic streaming and Staufen localization are convincing. Nonetheless, the authors do not address if tubulin polyglutamylation occurs in the germline, in the somatic tissues of the ovaries, or both. The fact that the female germline-specific TBA4 isotype is not glutamylated could argue in favor of glutamylation occurring mainly in the somatic cells. The lower amount of TBA4 compared to TBA1/3 detected by MS also reflects the high quantity of tubulin coming from follicle cells, which may mask glutamylation of TBA4.Since the contribution of kinesin 1 to the TTLL5 mutant phenotypes in oocytes or neurons has not been addressed, it is premature to conclude that these phenotypes are mediated by an effect of microtubules polyglutamylation on kinesin-based transport.The authors do not comment on the fact that a previously published study (Devambez et al. 2017) did not detect any effect of TTLL5 loss-of-function on vesicular transport. The discrepancies could be due to the experimental system or more likely to the nature of the TTLL5 alleles used in the present study which are catalytically-dead enzymes that may act as dominant negatives. It would be interesting to compare the different mutants in the same system and to rescue the transport phenotype with wild-type TTLL5, to firmly establish the role of TTLL5 in mitochondrial transport.The authors also report the presence of polyglycylated α-tubulin in ovaries, detected by mass spectrometry. This observation, which is unexpected given that the known *Drosophila* polyglycylases (TTLL3A/B) are not expressed in ovaries, is potentially interesting but should be confirmed by western blotting.It would be meaningful to analyze by immunostaining the distribution of polyglutamylated microtubules (MTs) in ovaries to determine if this modification occurs in follicle cells, oocytes, or both. One easy way to demonstrate that polyglutamylation occurs in the germline could be to express a GFP-tagged α-tubulin using the nos-gal4 driver and analyze by western blotting if this protein is polyglutamylated. This approach could also allow determining if the TUBA4 isotype can be polyglutamylated.Another approach could be to make TTLL5 mutant clones in the germline by FRT-mediated recombination and analyze their phenotype (cytoplasmic streaming and localization of Staufen). This would demonstrate that the phenotypes are not due to defects in follicle cells and would allow the analysis of the MT network in TTLL- oocyte. Does Staufen colocalize with glutamylated MTs in wild-type oocytes?

To illustrate that oocyte MTs can be polyglutamylated, we have now included a figure that shows that glutamylation of MTs is seen in oocytes, nurse cells, and follicle cells. This signal depends on *TTLL5,* indicating that *TTLL5* is indeed active in the germline (Figure 4). In S10B oocytes, the polyglutamylated MT signal is enriched in the cortical region. Because it has been described by different groups that cortical kinesin activity in this region drives the cytoplasmic streaming that helps to localize Staufen to the posterior, the cortical localization of TTLL5 (Figure S3) and the polyglutamylated signal (MTs) is consistent with the proposed role of TTLL5 and MT polyglutamylation.

The mRNA levels of the general α-tubulin *αTub84B* and *D* (*TBA1/3*) are expressed at “extremely high” levels in 0-2 hours old embryos (FlyBase data), which only contain mRNAs expressed in the maternal germline (i.e. in the ovary during oogenesis).

The existence of polyglycylated tubulin in the female germline should be confirmed by western blotting using specific antibodies that are available.

We have removed most of the work on glycylation because we cannot rule out that the identification is a mass spec artifact. In fact, it is very likely to be one, because the requested Western blot failed to show a proper signal (thank you for proposing this!). We thought we should point out this problem in the result section.

It is important to address the differences between the present study and the results from Devambez et al. Do the catalytically dead enzymes behave as dominant negative? Are the transport phenotypes observed in TTLL5E517A rescued by expression of Venus-TTLL5? Are they observed in TTLL5E517A/+ flies?

We used a mixture of *TTLL5^E517A^/+* and *Df(TTLL5)/+* as controls for our experiments. In these situations, we did not observe a dominant phenotype of the enzymatically dead mutant. In addition, we did not observe differences between the enzymatically dead mutant and the null in the localization of Staufen in oocytes.

However, in the cytoplasmic streaming assays, we did observe a higher penetrance (100% as opposed to 75%) and possibly a higher expressivity of the mutant phenotype in the enzymatically dead mutant (Figure 6). We bring this issue up in the new figure legend, but also point out that the numbers for the enzymatic dead mutant are too low to firmly conclude this. Comparing the null and enzymatically dead mutant in the two situations in the ovary that we have focused on, the mutant phenotype was clearly seen in both types of mutants, showing that this is a loss-of-function phenotype.

We interpret this to mean that the possibly slight dominant negative effect might affect mainly the strength (penetrance and possibly expressivity) of loss-of-function phenotypes but is unlikely to cause a neomorphic phenotype.

I have concerns regarding the signal detected by β-monoE antibody in ovaries. According to the initial publication (Bodakuntla et al. 2021), the β-monoE antibody recognizes glutamylated GEF motif only, but this sequence is found only in the tail of βTub85D isotype which is testis specific. Did the authors address the specificity of this new antibody in *Drosophila*? Did the author analyze glutamylation sites of βTubulins by mass spectrometry?

We removed the β-monoE results.

In the text, "tubulin isoforms" should be changed to "tubulin isotypes" since they are produced from different genes.

We have replaced all of them.

[Editors’ note: what follows is the authors’ response to the second round of review.]

The manuscript has been improved but there are some remaining issues that need to be addressed, as outlined below:The reviewers were split upon re-review of this work. There were concerns that the TTLL4 data were too preliminary and do not strengthen the paper and should be removed.

We agree and have removed the part related to TTLL4A.

In addition, all reviewers agreed in the discussion that the speculation about microtubule severing should be removed entirely. One reviewer pointed out that the authors use the example of spastin, but spastin-mediated severing is mainly affected by polyglutamylation on β-tubulin, which is not the case in this paper.

We have removed the severing part from the discussion.

The general consensus in the review discussion is that there is still a lack of experimental support for a direct effect of microtubule glutamylation on kinesin-1 activity. Therefore, the suggestion is for the authors to soften the claims and provide a sentence or two about how future in vitro studies will be necessary to determine if there is a direct effect on the motor itself or if there are effects on other MAPs, enzymes or processes that could produce the in vivo results.

We have softened the conclusion and added the requested discussion in the second but last paragraph of the discussion.

Reviewer #1 (Recommendations for the authors):The authors set out the examine the in vivo function of glutamylation of the C-terminal tails of microtubules. Tubulin glutamylation is widely observed but the precise tubulin isotypes that undergo this modification and the in vivo consequences of these changes are not well understood. The authors provide evidence that specific tubulin isotypes undergo this modification and that these events influence kinesin-1 function using well established model systems in oogenesis and in wing nerve axons. This work complements and extends previous studies of the function of glutamylation of microtubules in vivo. Strengths of the study include painstaking generation and characterization of several new alleles in the TTLL5 glutamylase. The use of multiple alleles strengthen the conclusions drawn. These reagents will also be valuable for other researchers. The conclusions are supported by quantitative analysis, including of several readouts of kinesin-1 activity. Whilst the authors do not address how polyglutamylation affects kinesin-1 function, the work will stimulate other efforts to address this issue.The authors have satisfactorily addressed most of my comments. In particular, the wing nerve analysis is much improved and has led to interesting conclusions. The revisions in response to the other reviewers have also led to significant improvements. Whilst the function of microtubule glutamylation has been assessed in other systems, the work represents a valuable addition and also generates and characterizes useful reagents for the community.However, I am still confused about some of the mass spectromery data. Whilst it seems clear from the rebuttal and revision that atub84B/D has a much higher frequency of glutamylation than atub67C, I do not understand which data support the conclusions in Figure 1. Is this a separate experiment to the one documented in Table 2 and S2? If so, the data should also be presented or the experiment leading to Figure 1 not mentioned (if the conclusions are fully supported by the later experiment). If all these data come from the same experiment, this issue should be clarified and the tables called out before Figure 1.The evidence supporting the conclusion that tub84B/D is differentially glutamylated vs tub67C would be easier to extract if the percentage modification and total number of peptides was additionally stated for both categories in the text.

All the MS data come from the same experiment. The proposed order is indeed better, and we have moved the original Table S2 up to the main manuscript as Table 2A and the previous Table 2 has now become Table 2B.

We now include supplementary tables (Supplementary Files 1b-d) to show the identified peptides on which the Tables 2A and 2B and Figure 1B are based. Supplementary Files 1b & 1c present a comprehensive list of unmodified and Glu-modified full-length primary Cterminal sequences of α-tubulins as obtained by the MS. These data were used to quantify the data and provide numbers and ratios for Tables 2A and 2B. Please note that we have conducted an additional search using the pipeline program MSGF and we found one single Glu-modified peptide in one of the *TTLL5* mutants, which had not been identified before. Therefore, we have made an adjustment to the ratios and numbers in Table 2B to accurately reflect this new finding. We mention that we do not know whether this is a sample contamination, a “memory effect” of the MS column, or caused by the residual activity of the *TTLL5^PBac^* (hypomorph?) or by another TTLL. Supplementary Files 1b-1d also show how these peptides were identified. We also reworked Figure 1B in a way that it now shows the different possible positions of the modifications with a good probability score and, superimposed on this, the very high scores obtained with the specialized programs indicated.

Reviewer #2 (Recommendations for the authors):In cells, microtubules acquire chemical modifications that are thought to affect the interactions between microtubules and microtubule-binding proteins. The authors address the question of what role microtubule glutamylation may have on microtubule function in cells by manipulating the enzyme that glutamylates microtubules and using *Drosophila* as a model.A strength of the manuscript is the authors' in vivo analysis of the TTLL5 enzyme that glutamylates microtubules. The mutations that were introduced into endogenous TTLL5 create a useful toolbox for the in vivo analysis of TTLL5 and microtubule glutamylation. Also, in general, the data are clearly presented and are consistent with the authors conclusions. Major weaknesses of the manuscript, however, are that (1) the authors' central model that microtubule glutamylation affects kinesin-1-dependent processes is supported by correlative data but not any functional tests,

We have made relevant adjustments as suggested by the editors.

and (2) the phenotypes resulting from the loss of microtubule glutamylation are quite modest.

It is indeed challenging to find strong phenotypes in *TTLL5* mutants, despite our extensive investigations across different tissues. However, evaluating the cellular functions of *TTLL5* and MT glutamylation is still meaningful. In fact, it is effects like this (and not lethality or sterility) that drive neurodegeneration and cancer.

As we have mentioned in the discussion, the fact that *TTLL5* mutations are not detrimental to survival makes this mutant a potentially favorable candidate for reversing the hyperglutamylation in neurodegenerative diseases. Furthermore, one of the goals of the study is to establish a useful but relatively simple in vivo model and methodology for further exploring the underlying mechanism of MT glutamylation and its cellular function. This we achieved.

This study is a nicely done characterization of a limited suite of TTLL5 phenotypes in an in vivo model (fly oocyte), and the authors' description of the effects of the loss of microtubule glutamylation will be of interest to researchers working in the field of microtubule post-translational modifications.A central challenge in the field of microtubule post-translational modifications (PTMs) is to determine the physiological significance of these PTMs and the mechanism by which they may act. The authors generate useful TTLL5 reagents and characterize the effects of disrupting microtubule glutamylation in oocytes (and neurons) in vivo, but these observations by themselves do not provide sufficient support to the authors' central model that microtubule glutamylation affects kinesin-1-mediated processes. The authors have generated new tools and information that will be useful to the field, but the modest phenotypes and lack of functional tests between TTLL5 and kinesin-1 limit the conceptual and scientific advance.Overall, this is a solid but limited characterization of TTLL5 mutants. My major concerns remain about the model connecting microtubule glutamylation and kinesin-1: this model is based on correlation and somewhat modest phenotypes. There is also quite a bit of speculation (the introduction of microtubule severing into a potential mechanism seems very speculative). Some of my original major concerns were not addressed in the revision. I do not support publication.

We have deleted the severing part in the discussion as suggested by the editors.

Upon reviewing the previous exchange, we acknowledge that we had not clearly addressed the following points from the previous round.

Have the authors tested for a genetic interaction between TTLL5 and kinesin-1 in oocytes and neurons? The kinesin-1 mutation that disrupts microtubule sliding and cytoplasmic streaming in oocytes (mutA mutation) would likely be quite helpful, too.

Thanks for the suggestion. We have not tested the genetic interaction between *TTLL5* and Kinesin-1. As suggested by the editor, we have cited other literature describing the connection between glutamylated MTs and kinesin-1. We have now also added a paragraph in the discussion with suggested future efforts that should bring further advances.

Zheng et al., discovered that different ER proteins interact through different proteins with different MT populations and that glutamylation changes also affect these interactions and localizations. It would have been beyond this paper to add similar studies from *Drosophila* ourselves, but we have included this point in the discussion.

3) The authors describe reports from mammalian models that increased glutamylation, rather than loss of glutamylation, may be disruptive. What are the effects of increasing TTLL5 in oocytes and/or neurons in flies?

We had also examined the effects of increasing TTLL5 in the female germline (Figures 3B, D, G) and its effect on ooplasmic streaming (Figure 6) and Khc distribution in the occyte (Figure 7). These results are still presented in the new version. We have not done analogous studies in the wing/neurons.

Reviewer #3 (Recommendations for the authors):Bao et al. investigate the function of Tubulin glutamylation in *Drosophila* oocyte and in mitochondria transport in adult nerve. Using Mass Spectrometry (MS), they detect glutamylation only on αTub84B/D C-terminal tails but not on the maternal αTub67C isotype. αTub84B/D are mainly monoglutamylated and polyglutamylation does not extend more than 3 glutamates. They identify TTLL5 as a major glutamylase during oogenesis, using previously described mutants as well as new point mutation alleles produced by CRISPR. In TTLL5 mutant backgrounds, cortical localization of Staufen at the posterior pole of S10 oocytes is more diffuse, and cytoplasmic streaming is abnormal. Significantly, these phenotypes are partially rescued by expression of a Venus-TTLL5 construct. The authors show that localization of Khc in S10 oocytes is also affected by loss-of-function as well as upregulation of TTLL5. Finally, they show that mitochondria transport in the wing nerve is affected in TTLL5 mutants background in which pausing time is decreased in anterograde but not in retrograde transport. Altogether these data suggest a link between microtubules glutamylation mediated by TTLL5 and kinesin1-dependent processes.In this revised version of the manuscript the authors took into account most criticisms of the reviewers and overall, the manuscript has been improved. Some data and conclusions that looked preliminary in the first version have been completely deleted. On the other hand, new data and quantifications, especially in Figures 6, 7 and 8, now strengthen the results and conclusions of the authors. Presentation of the results has also been improved. However, new data concerning the potential role of TTLL4A as a glutamylase in the ovary are preliminary and miss important controls. The relative contribution of TTLL4A and TTLL5 to αTub84B/D glutamylation is not explored.I have concerns about the data on TTLL4A. Indeed, the authors now explored the function TTLL4A (Figure S2). It is not clear why they initially supposed that TTLL4A could be involved in glycylation of αTubulin. The presence of glycylation in ovaries is not supported by strong data and the authors recognize that it could be due to an artifact of the MS procedure itself. Moreover, one cannot rule out that detection of low glycylation levels on αTubulin by MS is due to a contamination of the ovaries sample by sperm present in spermateca. Indeed, the polyglycylation signal is extremely strong on sperm axoneme. Analyzing the ovaries of virgin females would definitely prove that glycylation is present or not during oogenesis.

As suggested by the editors, we have removed the entire TTLL4A part. We still thank you for the useful tips and will take them into account for future experiments.

More importantly, there is not sufficient data to assure that TTLL4A is also involved in glutamylation of αTubulin. First, the mutants used are not characterized. At least RT-PCR is necessary to show that the expression level of TTLL4A is decreased in the mutant contexts used. Moreover, no rescue of the phenotype is presented. It seems important to note that the genomic deficiency used, which includes TTLL4, also deletes piwi and aubergine, two genes which have important roles during oogenesis. It is therefore necessary to rule out that the transposons insertions in the TTLL4 gene do not impact the expression of a nearby gene. In the western blots all genotypes should be probed with both GT335 and 1D5 as well as with a Tubulin antibody as loading control.Finally, the impact of TTLL4 depletion on tubulin glutamylation, as presented, is as strong as the impact of TTLL5 (Figure S2). How do the authors reconcile this result with their observation that no glutamylation is detected in TTLL5 mutant by MS? What is the relative contribution of TTLL4A and TTLL5 to αTub84B/D glutamylation? Using RNAi-mediated depletion of TTLL4A and TTLL5 would be a simple strategy to address these questions.Overall, the data with TTLL4 are too preliminary and do not strengthen the conclusions of the paper.

Thank you for your detailed feedback and suggestions. We were aware of the contradictory results and pointed this out already in the previous version. We agree with the points raised here and have deleted the TTLL4A part. Nevertheless, we would like to address some of them. As we now show in the new manuscript, an additional search of the same MS data revealed a single Glu-modified C-terminal peptide in the *TTLL5^oBAc^* mutant and we, therefore, have made an adjustment to the ratio and the number of Glu-MT in the *TTLL5^pBac^* mutant in Table 2B to accurately reflect these new findings. The origin of this peptide remains unclear as discussed in the paper and before in this response document. We also observed a faint band on the WB of the *TTLL5* mutants when incubating them with the GT335 antibody (Figure 3B). These faint bands could be due to antibody cross-reactivity, or the presence of a small amount of mono-Glu modification. In the latter case, we do not know whether this signal stems from a residual activity of this mutant, from a modified E in a more N-terminal region (see below), or a modification by another enzyme (e.g., TTLL4). To clarify this, we would need a much larger sample size and additional genetic material.

We can, however, mention that we ahve identified three times the following peptide in *TTLL5^pBAc^* mutants and once in the wild type.

REDLAALEKDYEE[E1]VGMDSGDGEGEGAEEY

This glutamylated E434 residue is outside the region we have focused on in this study, but it might be recognized by the mono-Glu antibody as well because it is also in an acidic environment.

In most figures, the annotation of the genotypes is misleading since TTLL5 mutants are used as heterozygous over a deficiency. This is also not mentioned in the main text but only in the figure legends. The genotypes are presented as TTLL5/- in figure 2 but not in other figures. The presentation should be consistent throughout the figures and clearly mentioned in the main text. Is there a stronger phenotype in TTLL5-/Def compared to TTLL5-/- homozygous? Why only hemizygous contexts were used?

In the Methods section we had stated that we analyze hemizygous animals. We have now added this to the main text, too, and also changed all the genotype annotations as suggested. The piggyBac and Minos transposon insertion chromosomes could potentially contain second-site mutations that become apparent when these chromosomes are made homozygous. We, therefore, routinely try to combine two unrelated chromosomes in order to avoid chasing the wrong mutant effect.

Figure 6 The number of oocytes (7) for the rescue experiment is too low to draw the conclusion that streaming is restored in the majority of oocytes.

We agree that this number is on the low side. However, each oocyte in the experiment was taken from different flies. Given that so far 5 out of 7 oocytes from 7 flies showed normal streaming, it means that the next 14 oocytes would ALL have to show abnormal streaming to bring the rescue down to the mutant level (in the *TTLL5^MiEx^* mutant, 5/21 were normal).